# International meta-analysis of PTSD genome-wide association studies identifies sex- and ancestry-specific genetic risk loci

Caroline M. Nievergelt ⬤ et al.[#]

The risk of posttraumatic stress disorder (PTSD) following trauma is heritable, but robust common variants have yet to be identified. In a multi-ethnic cohort including over 30,000 PTSD cases and 170,000 controls we conduct a genome-wide association study of PTSD. We demonstrate SNP-based heritability estimates of 5–20%, varying by sex. Three genome-wide significant loci are identified, 2 in European and 1 in African-ancestry analyses. Analyses stratified by sex implicate 3 additional loci in men. Along with other novel genes and non-coding RNAs, a Parkinson's disease gene involved in dopamine regulation, *PARK2*, is associated with PTSD. Finally, we demonstrate that polygenic risk for PTSD is significantly predictive of re-experiencing symptoms in the Million Veteran Program dataset, although specific loci did not replicate. These results demonstrate the role of genetic variation in the biology of risk for PTSD and highlight the necessity of conducting sex-stratified analyses and expanding GWAS beyond European ancestry populations.

---

[#]A full list of authors and their affiliations appears at the end of the paper.

Post-traumatic stress disorder (PTSD) is a commonly occurring mental health consequence of exposure to extreme, life threatening stress, and/or serious injury/harm. PTSD is frequently associated with the occurrence of comorbid mental disorders such as major depression[1] and other adverse health sequelae including type 2 diabetes and cardiovascular disease[2,3]. Given this high prevalence and impact, PTSD is a serious public health problem. An understanding of the biological mechanisms of risk for PTSD is therefore an important goal of research ultimately aimed at its prevention and mitigation[4,5].

Exposure to traumatic stress is, by definition, requisite for the development of PTSD, but individual susceptibility to PTSD (conditioned on trauma exposure) varies widely. Twin studies over the past two decades provide persuasive evidence for at least some genetic influence on PTSD risk[6,7], and the last decade has witnessed the beginnings of a concerted effort to detect specific genetic susceptibility variants for PTSD[8,9].

The Psychiatric Genomics Consortium—PTSD Group (PGC-PTSD) published results from a large GWAS on PTSD, involving a trans-ethnic sample of over 20,000 individuals, approximately 5000 (25%) of whom were cases[10]. With this limited sample size, no individual variants exceeded genome-wide significance; however, significant estimates of SNP heritability and genetic correlations between PTSD and other mental disorders such as schizophrenia were demonstrated for the first time.

It is apparent from previous PGC work on other mental disorders that sample size is paramount for GWAS to discern common genome-wide significant variants of small effect that are replicable[11]. Subsequent to the publication of data from the first freeze[10], the PGC-PTSD has continued to acquire additional PTSD cases and controls through partnerships with an expanding network of investigators, such that we now have accrued a sample size that has enabled us to turn the corner on genome-wide risk discovery. Presented here are the results of our latest GWA studies that include over 23,000 European and over 4000 African ancestry PTSD cases, now involving a total trans-ethnic sample of over 200,000 individuals. In achieving this sample size, we identify sex- and ancestry-specific findings. GWAS and gene-based analyses across our cohorts indicate genome-wide significant associations, involving genes related to dopamine and immune pathways. We show high genetic correlations between PTSD and related psychiatric disorders such as major depressive disorder, but present evidence that some of the identified loci are

likely specific to PTSD. In addition, we construct a highly significant polygenic risk score for PTSD, which is predictive of re-experiencing symptoms (REX), a core feature of PTSD, in the independent Million Veteran Program cohort[9].

## Results

**Meta-analysis strategy across ancestries and sex.** We report meta-analyses of GWAS from the PGC-PTSD Freeze 2 (PGC2), comprised of an ancestrally diverse group of 206,655 participants (including 32,428 cases) from 60 different PTSD studies, ranging from clinically deeply characterized, small patient groups to large cohorts with self-reported PTSD symptoms (Supplementary Data 1). Trauma exposure included both civilian and/or military events, often with pre-existing exposure to childhood trauma, and the majority of controls were trauma-exposed. First ancestry groups were consistently defined across studies (Supplementary Fig. 1). Primary GWAS were then performed separately in the three largest ancestry groups (European (EUA), African (AFA), and Native American Ancestry (AMA)), then meta-analyzed across studies and ancestry groups. Given the previously observed differences between male and female heritability estimates in PGC-PTSD Freeze 1[10], we also performed sex-stratified analyses. Quantile-quantile plots showed low inflation across analyses (Supplementary Fig. 2), which was mostly accounted for by polygenic SNP effects with little indication of residual population stratification (see Supplementary Note 1 for additional information).

**Heritability of PTSD.** We estimated heritability of PTSD in the EUA studies (Table 1) based on information captured by genotyping arrays ($h^2_{SNP}$) from summary statistics using LDSC[12]. Assuming a population prevalence of 30% after trauma exposure, overall $h^2_{SNP}$ in PGC2 was 0.05 on the liability scale ($P = 3.18 \times 10^{-8}$). However, female heritability was highly significant ($h^2_{SNP} = 0.10$, $P = 8.03 \times 10^{-11}$), while male heritability was not significantly different from zero ($h^2_{SNP} = 0.01$, $P = 0.63$).

We further examined sex differences in heritability in different subsets of the data: the PGC2 data without the UK Biobank (referred to as PGC1.5) and the UK Biobank (UKB) by itself, which comprises a large proportion of PGC2. Similar to the overall PGC2, heritability in PGC1.5 was high in women and not significant in men. In contrast, in the UKB, male heritability

**Table 1 Heritability estimates in subjects of European ancestry (EUA)**

| Sample | N | N | 10% prev | | 30% prev | | 50% prev | | |
|---|---|---|---|---|---|---|---|---|---|
| | Cases | Controls | $h^2_{SNP}$ | 95% CI | $h^2_{SNP}$ | 95% CI | $h^2_{SNP}$ | 95% CI | p-value |
| *All* | | | | | | | | | |
| PGC2 | 23,212 | 151,447 | 0.04 | 0.02–0.05 | 0.05 | 0.03–0.07 | 0.06 | 0.04–0.08 | $3.2 \times 10^{-8}$ |
| PGC1.5 | 12,823 | 35,648 | 0.03 | 0.01–0.06 | 0.05 | 0.01–0.08 | 0.05 | 0.01–0.09 | 0.011 |
| *Men* | | | | | | | | | |
| UKB | 10,389 | 115,799 | 0.13 | 0.1–0.15 | 0.17 | 0.14–0.21 | 0.19 | 0.15–0.23 | $2.1 \times 10^{-18}$ |
| PGC2 | 9908 | 75,605 | 0.01 | −0.02 to 0.03 | 0.01 | −0.03 to 0.05 | 0.01 | −0.03 to 0.05 | 0.63 |
| PGC1.5 | 6364 | 23,905 | 0.01 | −0.04 to 0.05 | 0.01 | −0.05 to 0.07 | 0.01 | −0.05 to 0.08 | 0.69 |
| UKB | 3544 | 51,700 | 0.11 | 0.04–0.17 | 0.15 | 0.05–0.24 | 0.16 | 0.05–0.26 | $1.4 \times 10^{-3}$ |
| *Women* | | | | | | | | | |
| PGC2 | 12,973 | 73,627 | 0.07 | 0.05–0.09 | 0.10 | 0.07–0.13 | 0.11 | 0.07–0.14 | $8.0 \times 10^{-11}$ |
| PGC1.5 | 6128 | 9528 | 0.15 | 0.08–0.22 | 0.21 | 0.11–0.31 | 0.23 | 0.12–0.33 | $2.7 \times 10^{-5}$ |
| UKB | 6845 | 64,099 | 0.14 | 0.1–0.18 | 0.19 | 0.13–0.25 | 0.21 | 0.14–0.27 | $2.0 \times 10^{-10}$ |

Estimates are calculated using LD-score regression (LDSC) at different population prevalences after trauma exposure for the combined PGC freeze 2 samples, and separately for PGC1.5 (without the UK biobank), the UK biobank, and for men and women. Number of SNPs ranges from 1,160,174 to 1,175,791
P-value is testing if $h^2_{SNP}$ is different from zero and applies to all prevalences
*PGC2* all European ancestry subjects of PGC freeze 2 (including the UK biobank), *PGC1.5* European ancestry subjects in the PGC1.5 EUA (not including the UK Biobank subjects), *UKB* UK Biobank European subjects, $h^2_{SNP}$ mean SNP-based heritability, *95% CI* 95% confidence interval, *prev* prevalence

**Table 2 Comparison of heritability between European (EUA) and African ancestry (AFA) studies**

| Sample | N | N | 10% prev | | 30% prev | | 50% prev | | |
|---|---|---|---|---|---|---|---|---|---|
| | Cases | Controls | $h^2_{SNP}$ | 95% CI | $h^2_{SNP}$ | 95% CI | $h^2_{SNP}$ | 95% CI | *p*-value |
| *All* | | | | | | | | | |
| EUA | 9354 | 25,175 | 0.04 | 0.02–0.06 | 0.05 | 0.02–0.08 | 0.05 | 0.02–0.08 | $1.3 \times 10^{-4}$ |
| AFA | 3163 | 9459 | 0.02 | −0.04 to 0.09 | 0.03 | −0.06 to 0.12 | 0.04 | −0.06 to 0.13 | 0.22744 |
| *Men* | | | | | | | | | |
| EUA | 4412 | 17,380 | 0.02 | −0.02 to 0.05 | 0.02 | −0.02 to 0.07 | 0.03 | −0.03 to 0.08 | 0.15951 |
| AFA | 1195 | 4361 | 0.02 | −0.14 to 0.18 | 0.03 | −0.2 to 0.25 | 0.03 | −0.21 to 0.27 | 0.41127 |
| *Women* | | | | | | | | | |
| EUA | 4689 | 5874 | 0.08 | 0.03–0.13 | 0.12 | 0.05–0.19 | 0.13 | 0.05–0.20 | $4.0 \times 10^{-4}$ |
| AFA | 1761 | 4435 | 0.12 | −0.01 to 0.25 | 0.17 | −0.01 to 0.35 | 0.18 | −0.01 to 0.38 | 0.028 |

Analyses are performed using GCTA in both sexes and for men and women separately and include all subjects used in the EUA and AFA GWAS with access to individual-level genotype data. Number of SNPs ranges from 4,071,335 to 4,863,146.
P-value is testing if $h^2_{SNP}$ is different from zero and applies to all prevalences
$h^2_{SNP}$ mean SNP-based heritability, 95% CI 95% confidence interval, prev prevalence

was significant ($h^2_{SNP} = 0.15$, $P = 1.38 \times 10^{-3}$) and not significantly different ($z = 0.23$, $P = 0.41$) from heritability in women ($h^2_{SNP} = 0.19$, $P = 2 \times 10^{-10}$). Sensitivity analyses in UKB using different case and control definitions further confirm these results (Supplementary Table 1 and Supplementary Note 1).

We also compared heritability across ancestries using the GCTA GREML method, which allows analysis of admixed populations when individual genotypes are available. GCTA estimates in the smaller EUA data remained similar to LDSC on the full data (Table 2). Heritability in AFA was comparable to estimates for EUA in the overall sample and when stratified by sex.

**Comparability of PGC2 studies and sex-specific analyses.** A comparison of heritability estimates in subsets of PGC2 studies stratified by sex, ancestry, and characteristics of study (i.e. PGC1.5 vs. the large UKB cohort) show significant estimates for PTSD in the range of 10–20% (Tables 1–2). This is with the notable exception of PGC1.5 males (in EUA and AFA analyses), which fail to show significant $h^2_{SNP}$, despite similar numbers of PTSD cases compared to PGC1.5 women. To further evaluate the comparability of PGC2 studies we estimated genetic correlations ($r_g$) between subsets with different characteristics (Supplementary Table 2).

As numerous small studies contribute to PGC1.5 (24 EUA studies with $N < 200$ cases, Supplementary Table 3), we first investigated the contribution of small studies to the overall genetic signal and genetic similarity to the larger PGC1.5 cohorts. The combined subset of small studies showed significant overall heritability ($h^2_{SNP} = 0.12$, $P = 0.046$) and close to significant genetic correlation with large studies ($r_g = 0.45$, $P = 0.08$), indicating a meaningful genetic contribution in aggregate.

Subsetting PGC1.5 and UKB by sex showed a high genetic correlation between women and men for UKB ($r_g = 1.03$, $P = 1.6 \times 10^{-5}$), but no significant genetic correlation between women and men in PGC1.5, which was expected, since $h^2_{SNP}$ in PGC1.5 men is not significant. Next, focusing on PGC1.5 women only, a comparison to UKB showed significant genetic correlations with the overall UKB ($r_g = 0.46$, $P = 0.0004$) and UKB women ($r_g = 0.46$, $P = 0.0008$), and a slightly lower, but marginally significant correlation with the smaller UKB male data ($r_g = 0.44$, $P = 0.052$).

Overall, these findings of significant heritability estimates for PTSD and moderate to high genetic correlations between most PGC2 subsets, including PGC1.5 to UKB ($r_g = 0.73$, $P = 0.0005$), are promising for GWAS in these data.

**GWAS in subjects of European and African ancestry.** Our largest PTSD meta-analysis in subjects of EUA (maximum number of subjects included in EUA meta-analyses: $N = 23,212$ cases, 151,447 controls, see Table 3 for details) identified two independent, genome-wide significant loci ($P < 5 \times 10^{-8}$), both mapping to chromosome 6, and sex-stratified analyses in men identified two additional loci (Fig. 1a, b, respectively). The smaller meta-analyses in AFA ($N = 4363$ cases, 10,976 controls) identified one genome-wide significant locus, and an additional locus was found in men when stratified by sex (Fig. 1c, d, respectively). No genome-wide significant associations were found in meta-analyses of EUA or AFA women (Supplementary Fig. 3).

Regional plots of the six genomic regions can be found in Supplementary Figs. 4–9. The six leading markers show odds ratios of 1.12–1.33 and no significant heterogeneity across studies (Table 3). This is supported by PM-plots (posterior probability that a SNP effect exists in a given study) showing a high consistency of effects among the studies predicted to have an effect[13] (Supplementary Figs. 10–15). A z-test on the effect sizes confirmed similar effects for men and women for the three leading variants in the joint-sex analyses, and significant sex-specificity for the three male hits identified in the sex-stratified analyses (Supplementary Table 4).

**Analyses across ancestry groups.** In order to study whether the genetic associations with PTSD vary across different ancestries, we first tried to replicate our six EUA and AFA top hits in the other main ancestry groups (EUA, AFA and AMA, respectively). No evidence of replication was found by directly comparing the six leading markers, nor by investigating the larger genomic regions harboring the signal (Supplementary Figs. 16–21). In addition, we did not identify any genome-wide significant hits by performing a trans-ethnic genome-wide meta-analysis across the six main ancestry groups ($N = 29,556$ cases and 166,145 controls) under fixed- and random-effect models (Supplementary Fig. 22).

While lack of replication of the 4 EUA hits may not be conclusive due to limited power in the smaller AFA and AMA data (Supplementary Tables 5–6), the 2 hits in AFA provided an opportunity for a more detailed analysis of ancestry effects. GWAS in the AFA subjects included standard PCs to control for average admixture across the genome. However, to precisely infer local ancestry across the genome of admixed subjects, we further implemented a local ancestry inference (LAI) pipeline (Supplementary Note 1 and Supplementary Fig. 23). We confirmed the AFA top hit rs115539978 to be specific to the African ancestral background (8% MAF on the African, and <1% MAF on the

**Table 3 Meta-analyses of European (EUA) and African (AFA) ancestry GWAS**

| Subjects | Variant | Chr | A1 | A1 freq | Beta | SE | OR | 95% CI | P-value | N cases | N controls | N effective | # sign. markers | I² | Q | N studies |
|---|---|---|---|---|---|---|---|---|---|---|---|---|---|---|---|---|
| *EUA* | | | | | | | | | | | | | | | | |
| All | rs34517852 | 6 | a | 0.34 | 0.110 | 0.02 | 1.12 | 1.08-1.16 | $3.1 \times 10^{-9}$ | 12,080 | 33,446 | 30,274* | 4 | 14.23 | 0.22 | 41 |
| All | rs9364611 | 6 | t | 0.13 | −0.124 | 0.02 | 0.88 | 0.85-0.92 | $4.3 \times 10^{-8}$ | 23,212 | 151,447 | 70,332 | 1 | 0.00 | 0.60 | 43 |
| Men | rs571848662 | 19 | t | 0.61 | −0.139 | 0.02 | 0.87 | 0.83-0.91 | $7.8 \times 10^{-9}$ | 6263 | 22,971 | 16,964* | 1 | 0.00 | 0.83 | 31 |
| Men | rs148757321 | 1 | ctgtg | 0.83 | 0.168 | 0.03 | 1.18 | 1.11-1.26 | $3.7 \times 10^{-8}$** | 6263 | 22,971 | 16,964* | 2 | 26.16 | 0.09 | 31 |
| *AFA* | | | | | | | | | | | | | | | | |
| All | rs115539978 | 13 | t | 0.07 | 0.284 | 0.05 | 1.33 | 1.20-1.47 | $2.7 \times 10^{-8}$ | 4363 | 10,976 | 11,322 | 10 | 13.95 | 0.28 | 21 |
| Men | rs142174523 | 6 | a | 0.30 | −0.277 | 0.05 | 0.76 | 0.69-0.84 | $4.3 \times 10^{-8}$** | 1782 | 5361 | 4702 | 1 | 0.00 | 0.76 | 13 |

Leading markers for genome-wide significant loci (at $p < 5 \times 10^{-8}$) in the overall and sex-stratified analyses are reported. The imputation information score ranges from 0.64 (rs34517852) to 1.16 (rs115539978), with a median score ranging from 0.89 to 0.97. *CHR* chromosome, *A1* allele 1 (coded allele), *A1 freq* A1 allele frequency, *SE* standard error, *OR* odds ratio, *I²* heterogeneity index, *Q* p-value for Cochran's Q statistic
*Not imputed in UK biobank
**Not genome-wide significant when adjusting for sex-split analyses ($p < 1.67 \times 10^{-8}$)

European and Native American backgrounds, respectively; (Supplementary Table 7). Conversely, LAI analyses of the male-specific hit indexed by rs142174523 showed no evidence for ancestry-specific effects that would explain the lack of replication in the EUA meta-analyses (Supplementary Table 8); however, the LD-structure in the MHC locus is complex[14].

**Integration with functional genomic data**. Functional mapping and annotation of the 6 GWAS hits using the FUMA pipeline conservatively predicted five genes *ZDHHC14*, *PARK2*, *KAZN*, *TMEM51-AS1* and *ZNF813* located in EUA risk loci, and five distinct genes *LINC02335*, *MIR5007*, *TUC338*, *LINC02571* and *HLA-B* in AFA risk loci (Table 4). In addition, gene-based analyses on 18,222 protein-coding genes based on the EUA and AFA GWAS summary data identified two additional gene-wide significant loci, represented by *SH3RF3* ($P = 4.28 \times 10^{-07}$) and *PODXL* ($P = 2.37 \times 10^{-06}$) in the EUA analysis. Gene-based analyses in AFA did not result in genome-wide significant loci. The biological function and potential psychiatric relevance of the 12 genes predicted by FUMA are detailed in the Supplementary Note 1 and discussed below.

We next performed gene-set analyses to understand implicated genes in the context of pathways and found four significant, Bonferroni-corrected gene sets (Supplementary Data 2). Of note, the two gene sets identified in the EUA data point towards a role for the immune system in PTSD. This is supported by a number of TNF-related genes summarized in a significant gene-set in AFA.

Annotation of variants in risk loci showed limited evidence of functionality (Table 4 and Supplementary Note 1). Most notably for the AFA top hit on chromosome 13, when testing for chromatin interactions using Hi-C data in neural progenitor cells, significant chromatin conformation interactions were observed between the risk region and a region ~1100 kb upstream harboring additional non-coding RNAs including LINC00458, hsa-mir-1297 and LINC00558 as well as a region approximately 820 kb downstream harboring the pseudogene *HNF4GP1* (Supplementary Fig. 24). eQTL analyses did not show significant associations with gene expression. However, the lack of functional data for this region may be explained by the African ancestry specificity of the GWAS findings since databases available within the FUMA framework, including GTEx and BrainSpan for eQTL analyses, are predominantly based on European populations.

Thus, we expanded our analyses for the AFA top hit through cell culture experiments in lymphoblastoid cell lines (LCLs) from African subjects (see Methods and Supplementary Note 1). We show evidence that the African-ancestry-specific SNP rs115539978 seems to capture a genomic region that may influence the expression of non-coding RNAs from this PTSD risk locus in response to increased glucocorticoid receptor signaling, thus linking this African-specific genetic variant to stress response and non-coding RNA expression (Supplementary Fig. 25).

We further characterized the AFA signal (rs115539978) using psychophysiology and imaging datasets available through the Grady Trauma Project (GTPC) and found evidence that this lead SNP captures a genomic region that is also associated with increased amygdala volume and fear psychophysiology in a traumatized population (Supplementary Note 1 and Supplementary Fig. 26).

**Replication of findings in the external MVP cohort**. In an attempt to replicate our genomics findings in an adequately-powered external study we used the large MVP cohort, including 146,660 EUA and 19,983 AFA participants assessed for

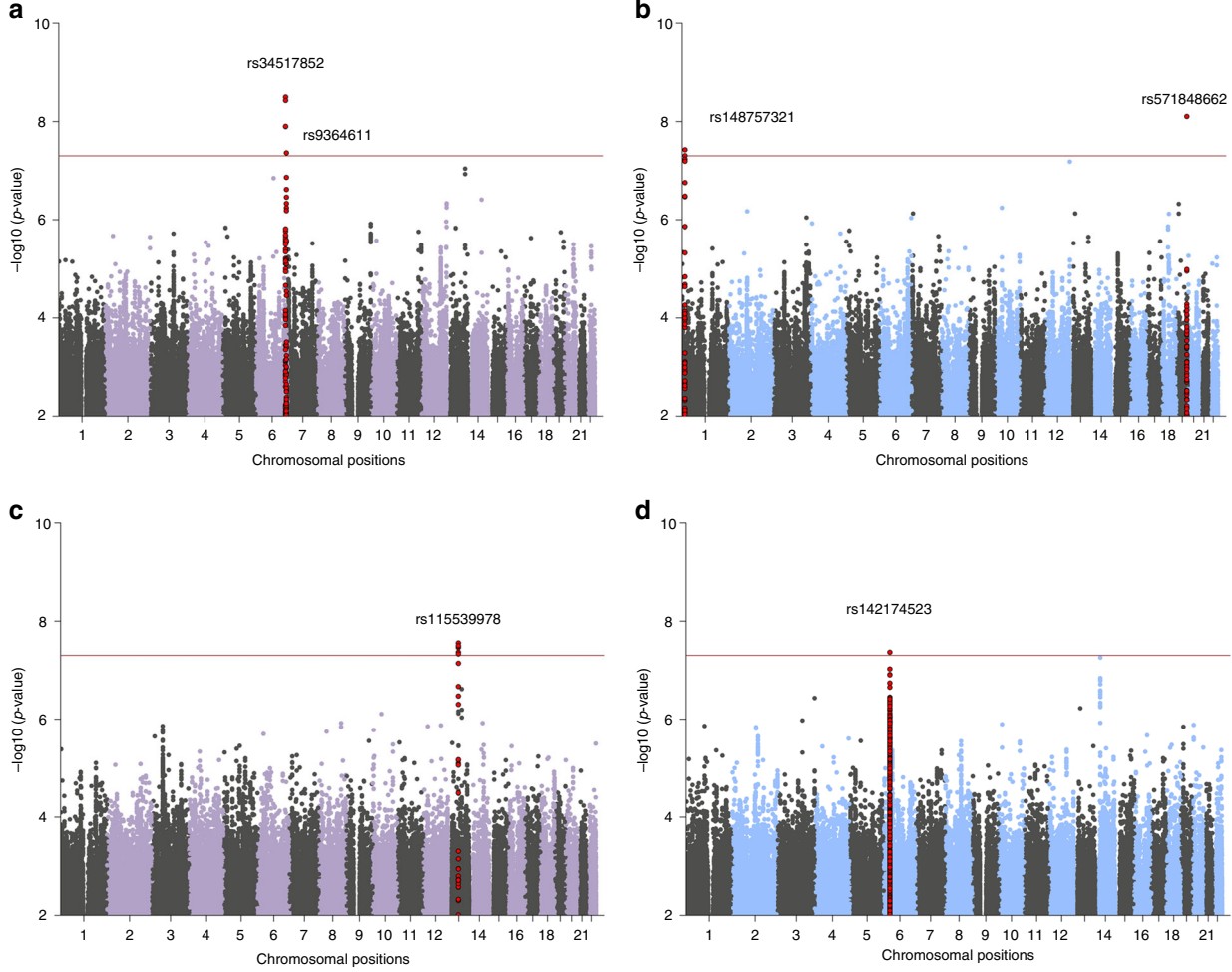

**Fig. 1** Manhattan plots from meta-analyses of PTSD GWAS, showing the top variants in six independent genome-wide significant loci. Results are shown for subjects of European (EUA; **a**) and African ancestry (AFA; **c**), and for sex-stratified analyses in EUA men (**b**) and AFA men (**d**), respectively. Sex-stratified analyses for women were not significant (Supplementary Fig. 4). The red line represents genome-wide significance at $P < 5 \times 10^{-8}$. Note: rs148757321 and rs142174523 do not remain significant after Bonferroni-adjustment for sex-stratified analyses (at $P < 1.67 \times 10^{-8}$)

re-experiencing symptoms (REX), a core feature of PTSD[9]. We first compared the genetic signals between PGC2 PTSD and MVP REX EUA and found a highly significant genetic correlation ($r_g = 0.80$, SE = 0.096, $P = 2.85 \times 10^{-17}$). No evidence of replication was found for the six leading PTSD markers identified in PGC2 EUA and AFA GWAS for the MVP REX-specific symptoms (Supplementary Table 9). However, two of these markers were not directly genotyped and had to be assessed by proxy markers in only moderately high (~75%) LD, and sex-stratified analyses were not available for MVP.

**Polygenic risk scores (PRS) for PTSD.** PRS generated from well-powered GWAS have recently become a tool of high relevance for polygenic disorders and traits (e.g. ref. [15,16]). We assessed the predictive value of PRS for PTSD, using our largest cohort, the UKB, as a training sample (Fig. 2a). Our analyses were strongest at a p-value threshold $P_T = 0.3$ and showed a highly significant increase in odds to develop PTSD across PRS quintiles in the PGC1.5 EUA target sample, with a variance explained on the liability scale of $r^2 = 0.0015$ (likelihood ratio test $P = 5.44 \times 10^{-7}$). Analyses within the UKB show even stronger PRS predictability, with the highest OR for UKB men with a PRS trained on UKB women, reaching an OR of 1.39 in the 5th quintile, with an overall variance explained of $r^2 = 0.012$ ($P = 4.19 \times 10^{-10}$).

We also tested the overall PGC2 PTSD PRS in the external MVP replication cohort, using REX as the target for predictions. PRS predictions were strongest at $P_T = 0.3$ and highly significant (likelihood ratio test $P = 5.4 \times 10^{-62}$, Supplementary Fig. 27). Mean REX symptoms in MVP EUA participants was 8.48 (4.59 SD), and participants in the 5th quintile of genetic risk had significantly higher REX scores than subjects in the 1st quintile (beta = 0.58, $P = 1.41 \times 10^{-48}$; Fig. 2b).

**Genetic correlation of PTSD with other traits and disorders.** Analysis of shared heritability across common disorders of the brain[17] and specific genetic correlations of psychiatric disorders with cognitive, anthropomorphic and behavioural measures[10,18–20] has been facilitated greatly by the development of a centralized database of GWAS results including a web interface for LDSC (LD Hub[21]). We estimated pairwise genetic correlations ($r_g$) between PTSD and 235 disorders/traits and found 21 significant correlations after conservative Bonferroni correction (Fig. 3, panel A and Supplementary Data 3). Genetic variation associated with PTSD was positively correlated with PRS from other psychiatric traits including depressive symptoms, schizophrenia and neuroticism, as well as epidemiologically comorbid traits such as insomnia, smoking behavior, asthma, hip-waist ratio and coronary artery disease. In contrast, negative $r_g$ with PTSD include subjective

**Table 4 Functional mapping and annotation of GWAS meta-analyses in the European and African ancestry data**

| Group | GWAS hit lead variant | #SNPs in LD (r² > 0.6) | genomic coordinate risk locus (hg19) | predicted genes in risk locus | SNPs in LD with CADD scores > 12.37 | SNPs in LD with RegulomeDB scores < 5 | Chromatin state analysis (Roadmap Epigenomics) in neuronal cell lines/tissues[a] | eQTL | Hi-C in 3 neuronal tissue/cell line datasets, GSE87112 |
|---|---|---|---|---|---|---|---|---|---|
| **EUA** | | | | | | | | | |
| All | rs34517852 | 20 | chr6: 157,780,424–157,801,753 | ZDHHC14 (upstream of TSS) | rs35262389 = 15.28 | rs9348095 = 1 (TSS site) | Transcriptional active chromatin at TSS | None | Yes, with downstream elements |
| | rs9364611 | 12 | chr6: 162,157,139–162,168,506 | PARK2 (intronic) | None | None | Overall quiecent, some enhancer function | None | Yes, intronic within the same intron |
| Men | rs571848662 | 5 | chr19: 53,988,841–53,990,834 | ZNF813 | None | None | Weak transcription | None | None |
| | rs148757321 | 10 | chr1: 15,427,801–15,449,791 | KAZN and TMEM51-AS1 | None | None | | None | Significant interaction with regions further upstream of KAZN |
| **AFR** | | | | | | | | | |
| All | rs115539978 | 61 | chr13: 55,652,129–55,759,209 | LINCO2335, MIR5007, TUC338 | None | None | Overall silenced chromatin (score of 15), some SNPs map to loci with weak transcription or enhancer function | None | Interaction between risk locus and upstream region harboring LINC00458, MIR1297, and LINC00558 as well as downstream region harboring HNF4GP1 |
| Men | rs142174523 | 237 | chr6: 31,257,622–31,319,815 (MHC locus) | LINCO2571, HLA-B | None | many[b] | Overall PolyComb repressed chromatin, heterochromatin | 16[c] | None |

GWAS in females of European and African ancestry did not identify genome-wide significant hits
TSS transcriptional start site, eQTL expression quantitative trait locus/cell lines from CommonMind Consortium, BRAINEAC or GTEx v7)
[a]In neuronal cell lines/tissues E053, E054, E067, E068, E069, E070, E071, E072, E073, E074, E081, E082, E125
[b]See http://fuma.ctglab.nl/browse 30–32 for a visualization of these results
[c]ATP6V1G2, C4A, C4B, CCHCR1, CYP21A1P, DDR1, HCG27, HLA-B, HLA-C, MICB, NOTCH4, POU5F1, PSORS1C3, SKIV2L, VARS and VARS2

well-being, education, and a strong correlation with parents' age at death ($r_g = -0.70$). Significant positive correlations were also found for reproductive traits such as the number of children ever born, and, as previously reported for women[22], PTSD was negatively correlated with age at first birth.

With the notable exception of asthma, our findings on PTSD correspond closely with genetic correlations between these traits and other psychiatric disorders such as MDD[18], SCZ[23], BPD[19] and ADHD[20] (Fig. 3, panels B-E). These findings are not surprising, as pleiotropic effects (i.e. SNPs impacting multiple traits) have been widely reported for psychiatric disorders.

In order to test to what degree genome-wide significant findings from our GWAS meta-analyses were specific to PTSD rather than driven by correlated traits as identified above, we adjusted the top hits from our analyses for the effects of genetically correlated psychiatric traits. Since the strongest correlations were found between PTSD and depressive symptoms ($r_g = 0.80$) and MDD ($r_g = 0.62$), summary statistics from PGC MDD[18], as well as MDD plus BPD and SCZ, were included in the analyses. Using a recently implemented method applicable to external GWAS summary data (mtCOJO[24]) to approximate an analysis where these traits are regressed out, we found that effect sizes for the four EUA top hits were not markedly reduced when adjusted for the effects of MDD, or all three psychiatric traits tested simultaneously (Supplementary Tables 10, 11). These findings indicate that the genetic variants identified here are specific to PTSD when tested in the context of the three psychiatric disorders genetically most significantly correlated with PTSD.

## Discussion

PTSD is a common and debilitating condition influenced by genetic factors, yet common genetic risk variants for PTSD have not been robustly identified. The PGC-PTSD combined data from 60 multi-ethnic cohorts (PGC1.5) and the UK Biobank (PGC2) achieved a sample size of 206,655 participants with 32,428 PTSD cases, over ten times that of any previous analysis[10,25]. As has been demonstrated in GWAS of SCZ[23], BPD[26], and recently in MDD[18,27] and ADHD[20], sample size is critical to produce robust genome-wide significant hits that inform foundational knowledge on the neurobiology of complex psychiatric conditions. These results show this is also true of PTSD. This increased power has led us to draw several major conclusions.

First, our genetic findings squarely place PTSD among the other psychiatric disorders in terms of heritability and genetic relationship with other disorders. While this statement may seem obvious to some, there remains debate about whether PTSD is entirely a social construction[28]. We found substantial SNP-based heritability (i.e. phenotypic variation explained by genetic differences) at 5–20%, similar to that for major depression[18] across methods, studies and ancestries. The heritability results and pattern of genetic correlations are also consistent with our initial findings[10] and with those from twin studies. PTSD shares common variant risk with other psychiatric disorders, which show substantial sharing of common variant risk with one another[29]. PTSD was most significantly (genetically) correlated with major depression, but also with schizophrenia, both of which have genome-wide significant loci implicated in brain function.

Second, our GWAS analyses identified several genetic loci not previously associated with PTSD. These loci pointed to a number of different target genes that merit further investigation. With *PARK2*, there is a posited role of dopaminergic systems in PTSD. The dopaminergic system has a critical role in fear conditioning which is important in the development and maintenance of PTSD[30]. There is also epidemiological evidence for association of

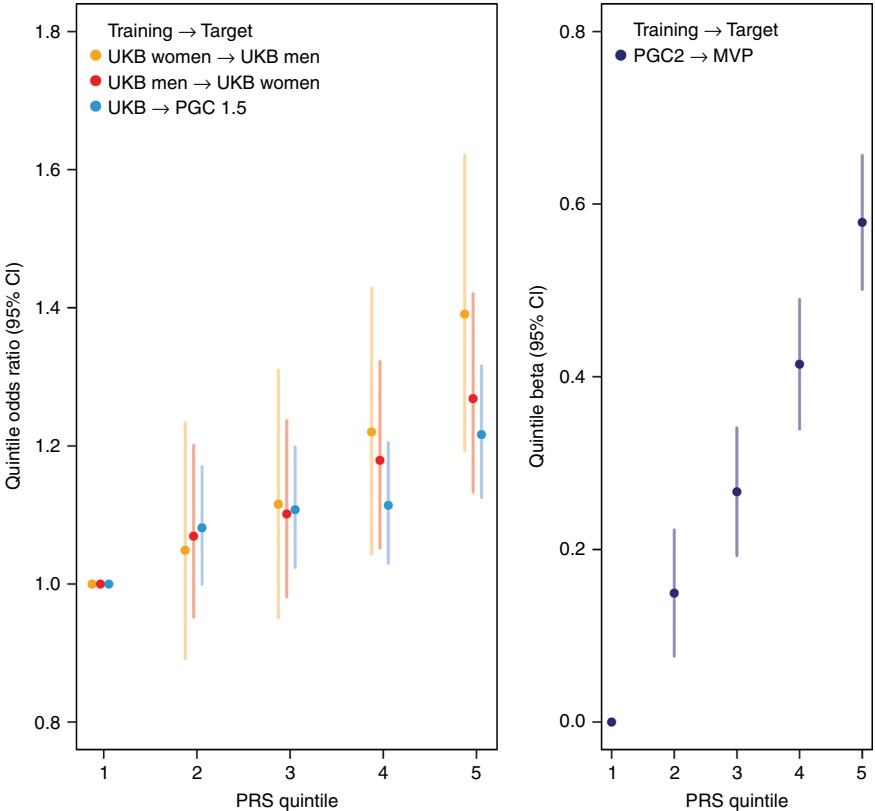

**Fig. 2** Genetic risk score (PRS) predictions for PTSD. **a** Using PTSD subjects from the UK Biobank (UKB) as discovery sample, odds ratios (OR) for PTSD per PRS quintile relative to the first quintile show a significant increase in different PGC PTSD target samples. For example, UKB men in the 5th quintile have 40% higher odds to develop PTSD than UKB men in the lowest quintile, when using women from the same population as a training set. **b** PRS predictions of re-experiencing symptoms in the external replication cohort from the Million Veteran Program (MVP) using the overall PGC2 as discovery sample show a highly significant increase in PTSD re-experiencing symptoms per PRS quintile. Sample sizes in different training and target sets include: UKB women: 6845 PTSD, 64,099 controls; UKB men: 3,544 PTSD, 51,700 controls; UKB: 10,389 PTSD, 115,799 controls; PGC1.5: 10,213 PTSD, 27,445 controls; PGC2: 23,212 PTSD, 151,447 controls; MVP: 146,660 participants with re-experiencing symptoms assessments. Analyses include only subjects of European ancestry

Parkinson Disease and PTSD[31–33]. *PODXL* is involved in neural development and synapse formation[34], *SH3RF3* is associated with neurocognition and dementia[35,36], *ZDHHC14* is associated with regulation of β-adrenergic receptors[37,38], and *KAZN* is expressed in the brain[39], where it has been found to be underexpressed in parvalbumin neurons of the superior temporal cortex of schizophrenia cases[40] and overexpressed in the substantia nigra of Parkinson's cases[41]. Finally, the *HLA-B* complex may be related through the known role of immunity and inflammation in stress-related disorders[42,43]. Less is known about the role of the identified RNAs *LINC02335*, *MIR5007*, *TUC338* and *LINC02571* in regards to the biology of PTSD. However, preliminary work from our group including imaging and psychophysiology highlights the value of deep phenotyping in conducting functional investigations into the meaning of GWAS findings[8]. Extensive follow-up work is needed to replicate our findings and to determine the function of identified genes and their relationship to putative pathological processes. For example, in SCZ the MHC locus is now thought to influence risk, in part, through pruning of synapses using immune machinery rather than through classical immune pathways[44]. These ancestry-specific results are preliminary, and even larger PTSD GWAS will facilitate the identification of plausible neurobiological targets for PTSD.

Third, our results also illustrate that there may be genetic contributions to the well-documented association between PTSD and dysregulation in inflammatory and immune processes[45]. It has been widely recognized that PTSD is associated with a broad range of adverse physical health conditions over the life course ranging from type-2-diabetes and cardiovascular disease to dementia and rheumatoid arthritis[46,47]. Less is known about the biological mechanisms driving the relationship between PTSD and these outcomes. Our genetic correlation analyses may provide some initial clues for further investigation. For example, we found a high genetic correlation ($r_g = 0.49$, $P = 0.0002$) between PTSD and asthma. Our subsequent gene-set and pathway analyses provide some clues further implicating the immune system. Of note, these genetic results converge with evidence from epidemiologic cohort studies documenting the role of stress-related disorders such as PTSD in autoimmune diseases[48], case-control studies showing elevations of immune-related biomarkers in women with PTSD[49], and epigenetic studies pointing to the role of the immune system in PTSD etiology[50,51]. Further work is needed to determine whether PTSD has genetic overlap with immune disorders broadly and the causal direction between disorders. At minimum, the emerging genetic evidence presented here suggests that association between PTSD and health conditions may, in some cases, have some genetic origin.

Fourth, PGC-PTSD is distinct in relation to current genomics consortia due to its high proportion of data from participants of diverse ancestries. For example, a recent review found that only three percent of all samples in genetic studies were from African ancestry[52]. This contrasts sharply with the 10% of AFA

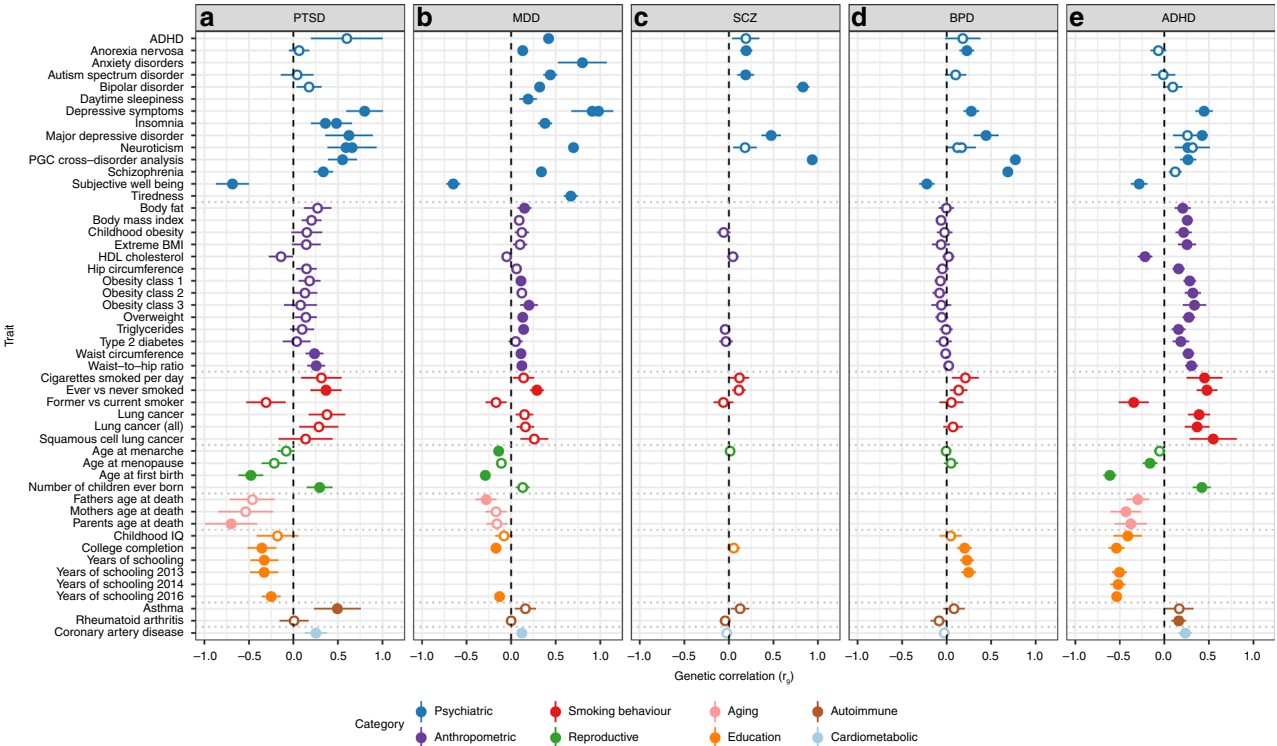

**Fig. 3** Commonality of genetic correlations between PTSD and other psychiatric disorders and traits with GWAS summary statistics on LD Hub. Psychiatric traits include **a** PTSD, **b** MDD, **c** SCZ, **d** BPD and **e** ADHD and their genetic correlations with traits from psychiatric, anthropomorphic, smoking behavior, reproductive, aging, education, autoimmune and cardiometabolic categories. Only traits with at least one significant correlation with the 5 psychiatric disorders are shown. Error bars indicate 95% confidence limits. Solid points indicate significant correlation after Bonferroni correction. The total number of correlations tested were 235 for PTSD, 221 for MDD, 172 for SCZ, 196 for BPD and 219 for ADHD

participants in our consortium. We have the first heritability estimates for PTSD in African ancestry: they are similar to EUA, highly significant in women and lower in men. Our GWAS in subjects of African ancestry indicated at least one ancestry-specific locus using local ancestry methods developed for this analysis. We note the sample size in the AFA analysis has only about 15,000 participants, which is small and under-powered, increasing the chance for false positives. However, other work has shown that genetic studies of underrepresented populations afford the opportunity to discover novel loci that are invariant in European populations[53]. As others have noted, there are major limitations in our knowledge of the genetic and environmental risk architecture of psychiatric disorders in persons of African descent[54]. Our findings provide further evidence of the need to invest in research that includes diverse ancestral populations, to expand reference data, and to continue to develop methods to analyze data from such populations. Until such an investment is made, we are limited in our ability to understand biological mechanisms, predict genetic risk[55], and produce optimal treatments for non-European populations. African genomes are characterized by shorter haplotype blocks and contain many millions more variants per individual than populations outside Africa[56]. Further, including data from African populations in genetic studies of PTSD and other neuropsychiatric disorders may accelerate genetic discovery and could be useful for fine mapping disease causing alleles[57].

Fifth, although PTSD heritability remained relatively stable across methods, studies, and ancestries, sex differences in heritability were observed in the overall cohort analyses as well as in the AFA analyses[10]. It is of note that the sex differences in heritability were not evident in the UK Biobank data, which we

hypothesize is due to differences between the PGC1.5 cohorts and the UK Biobank. PTSD is conditional on trauma exposure, which is also highly heterogeneous across individuals and populations[58]. Unlike PGC1.5, the UK Biobank cohort is comprised of few to no subjects who experienced military-related trauma. In contrast, a substantial proportion of men in the PGC1.5 cohorts were from military cohorts, while virtually all women were civilians. The environmental experiences (e.g. military versus civilian) and index traumatic events leading to PTSD in male subjects versus female subjects could explain observed lower heritability estimates in males in the PGC1.5 cohorts. In future work, we aim to investigate this empirically by pooling detailed trauma and PTSD phenotypic information on both males and females and by modeling the effects of measurement variability on heritability estimates. Future work could also aim to increase samples of civilian men and military women to allow for analyses stratified by military trauma and sex.

Lastly, we report a significant polygenic risk score for PTSD, which also significantly predicts re-experiencing symptoms in independent data from the MVP[9]. However, larger sample sizes are needed to achieve sensitivity and specificity at levels of clinical utility[16]. In the future, polygenic risk may eventually be useful in algorithms developed to identify vulnerable persons after exposure to trauma. PTSD is one of the most theoretically preventable mental disorders, as many people exposed to trauma come to clinical attention in first response settings such as emergency rooms, intensive care units, and trauma centers. Controlled clinical trials show that PTSD risk can be significantly reduced by early preventive interventions[59,60]. However, these interventions have nontrivial costs, making it infeasible to offer them to all persons exposed to trauma, given that only a small minority goes

on to develop PTSD[61,62]. They are also unnecessary for many survivors who recover spontaneously[59]. To be cost-effective, risk prediction rules are needed to identify which exposed persons are at high risk of PTSD. Such risk prediction tools have been developed[63,64], but none to date has included polygenic risk as a predictor.

These findings advance our understanding of the genetic basis of PTSD, but they also demonstate that PGC-PTSD remains under-powered for the detection of most risk loci and associated pathways. PTSD is similar to major depression in both prevalence (among trauma-exposed persons) and in heritability. There are now 100 genome-wide significant signals for major depression; notably, that level of discovery required 246,363 cases and 561,190 controls[27]. Other limitations include the treatment of PTSD as a binary disorder in our analysis. Extensive epidemiologic work has shown that subthreshold PTSD is highly prevalent and debilitating[65,66]. In our analysis, persons with subthreshold PTSD are classified as controls, which would likely reduce our power to find genetic associations. In future work, we will consider PTSD as a continuous phenotype as well as examine clusters of PTSD symptoms, which are more homogeneous. Of note, Gelernter et al. (2019) recently found multiple genome-wide significant loci for re-experiencing symptoms, which is the cluster of symptoms most unique to PTSD, in data from over 100,000 veterans in the Million Veteran Program[9]. Finally, we used mostly unscreened controls, but controls carefully screened for trauma may increase power since trauma is required for a PTSD diagnosis. In addition to increasing sample size, we aim in the future to also pool item-level phenotypic data from our cohorts in order to address these limitations.

Advances in understanding the genomic architecture of PTSD are critical for understanding the pathophysiology of this debilitating syndrome and to developing novel biologically-based treatment approaches. The current data from a PTSD GWAS in >195,000 participants advances our understanding of the genetic underpinnings of PTSD and trauma-related disorders.

## Methods

**Participating studies**. The PGC-PTSD Freeze 2 dataset (PGC2) includes 60 ancestrally diverse studies from Europe, Africa and the Americas. Of these, 12 were already included in Freeze 1[10]. Study details and demographics can be found in Supplementary Data 1. PTSD assessment was based either on lifetime (where possible) or current PTSD (i.e. including participants with a potential lifetime PTSD diagnosis as controls), and PTSD diagnosis was established using various instruments and different versions of the DSM (DSM-III-R, DSM-IV, DSM-5). For GWAS analyses, all studies provided PTSD case status as determined using standard criteria and control subjects not meeting the PTSD diagnostic criteria (see Supplementary Data 1 for additional exclusion criteria). The majority of controls was trauma-exposed. A detailed description of the studies included is presented in Supplementary Methods. We have complied with relevant ethical regulations for work with human subjects. All subjects provided written informed consent and studies were approved by the relevant institutional review boards and the UCSD IRB (protocol #16097×).

**Data assimilation**. Subjects were genotyped on a range of Illumina genotyping arrays (exception: UKB was genotyped on the Affymetrix Axiom array). At the time of analysis, direct access to individual-level genotypes was permitted for 65,555 subjects. For these, pre-QC'ed genotype data were deposited on the LISA server for central data processing and analysis, using the standard PGC pipelines (https://sites.google.com/a/broadinstitute.org/ricopili/) and (https://github.com/orgs/Nealelab/teams/ricopili). Studies with data sharing restrictions (eight studies, $N = 137,114$ subjects) performed analyses off site using identical pipelines unless otherwise indicated (Supplementary Data 1). Such studies then shared summary results for meta-analyses.

**Global ancestry determination**. To determine consistent global ancestry estimates across studies, each subject was run through a standardized pipeline, based on SNPweights[67] of 10,000 ancestry informative markers genotyped in a reference panel including 2911 unique subjects from 71 diverse populations and six continental groups ($K = 6$)[68] (https://github.com/nievergeltlab/global_ancestry). Pre-QC genotypes were used for these analyses.

For the present GWA studies, subjects were placed into three large, homogeneous groupings, using previously established cut-offs (Supplementary Table 12): European and European Americans (EUA; subjects with ≥90% European ancestry), African and African-Americans (AFA; subjects with ≥5% African ancestry, <90% European ancestry, <5% East Asian, Native American, Oceanian, and Central-South Asian ancestry; and subjects with ≥50% African ancestry, <5% Native American, Oceanian, and <1% Asian ancestry), and Latinos (AMA; subjects with ≥5% Native American ancestry, <90% European, <5% African, East Asian, Oceanian, and Central-South Asian ancestry). Native Americans (subjects with ≥60% Native American ancestry, <20% East Asian, <15% Central-South Asian, and <5% African and Oceanian ancestry) were grouped together with AMA. All other subjects were excluded from the current analyses ($N = 6,740$). Supplementary Fig. 1 shows the ancestry grouping used for GWAS of 69,484 subjects for which individual-level genotype data was available to the PGC. The ancestry pipeline was shared with external sites in order to ensure consistency in ancestry calling across cohorts.

**Genotype quality control**. The standard PGC pipeline RICOPILI was used to perform QC, but modifications were made to allow for ancestrally diverse data. In the modified pipeline, each dataset was processed separately, including subjects of all ancestries. Sample exclusion criteria: using SNPs with call rates >95%, samples were excluded with call rates <98%, deviation from expected inbreeding coefficient ($f_{het} < -0.2$ or >0.2), or a sex discrepancy between reported and estimated sex based on inbreeding coefficients calculated from SNPs on X chromosomes. Marker exclusion criteria: SNPs were excluded for call rates <98%, a > 2% difference in missing genotypes between cases and controls, or being monomorphic. Hardy-Weinberg equilibrium (HWE): the modified pipeline identified the largest homogenous ancestry group in the data, identified SNPs with a HWE $P$-value $< 1 \times 10^{-6}$ in controls, and excluded these SNPs in all subjects of the specific datasets, irrespective of ancestry.

**Relatedness within studies**. Within-study relatedness was estimated using the IBS function in PLINK 1.9[69]. From each pair with relatedness $\hat{\pi} > 0.2$, one individual was removed from further analysis, retaining cases where possible.

**Calculation of principal components (PC's) for GWAS**. For each dataset, unrelated subjects were subset into the three ancestry groups (EUA, AFA, AMA; Supplementary Tables 3, 5, 6) for analysis. SNPs were excluded that had a MAF <5%, HWE $P > 1 \times 10^{-3}$, call rate <98%, were ambiguous (A/T, G/C), or due to being located in the MHC region (chr. 6, 25–35 MB) or chromosome 8 inversion (chr. 8, 7–13 MB). SNPs were pairwise LD-pruned ($r^2 > 0.2$) and a random set of 100 K markers was used for each subset to calculate PC's based on the smartPCA algorithm in EIGENSTRAT[70].

**Imputation**. Imputation was based on the 1000 Genomes phase 3 data (1KGP phase 3[71]). Any dataset using a human genome assembly version prior to GRCh37 (hg19) was lifted over to GRCh37 (hg19). SNP alignment proceeded as follows: for each dataset, SNPs were aligned to the same strand as the 1KGP phase 3 data. For ambiguous markers, the largest ancestry group was used to calculate allele frequencies and only SNPs with MAF <40% and ≤15% difference between matching 1KGP phase 3 ancestry data were retained. Pre-phasing was performed using default settings in SHAPEIT2 v2.r837[72] without reference subjects, and phasing was done in 3 megabase (MB) blocks, where an additional 1 MB of buffer was added to either end of the block. Haplotypes were then imputed using default settings in IMPUTE2 v2.2.2[73], with 1KGP phase 3 reference data and genetic map, a 1 MB buffer, and effective population size set to 20,000. RICOPILI default filters for MAF and Info were removed since analyses were run across ancestry groups at this step. Imputed datasets were deposited with the PGC DAC and are available for approved requests.

**Main GWAS**. The analysis strategy for the main association analyses is shown in Supplementary Tables 3, 5 and 6. Analyses were performed separately for each study and ancestry group, unless otherwise indicated. The minimum number of subjects per analysis unit was set at 50 cases and 50 controls, or a total of at least 200 subjects, and subsets of smaller size were excluded. Smaller studies of similar composition were genotyped jointly in preparation for joint analyses (e.g. PSY1, PSY3). For studies with unrelated subjects, imputed SNP dosages were tested for association with PTSD under an additive model using logistic regression in PLINK 1.9, including the first five PC's as covariates. For family and twin studies (VETSA, QIMR), analyses were performed using linear mixed models in GEMMA v0.96[74], including a genetic relatedness matrix (GRM) as a random effect to account for population structure and relatedness, and the first five PC's as covariates. The UKB data (UKB) were analyzed with BGenie v1.2 (https://www.biorxiv.org/content/early/2017/07/20/166298) using a linear regression with 6 PC's, and batch and center indicator variables as covariates (see Supplementary Methods for additional details). In addition, all GWAS analyses were also performed stratified by sex.

**Meta-analyses.** Summary statistics on the linear scale (from GEMMA and BGenie) were converted to a logistic scale prior to meta-analysis (for formula see[75]). Within each dataset and ancestry group, summary statistics were filtered to MAF ≥1% and PLINK INFO score ≥0.6. Meta-analyses across studies were performed within each of the three ancestry groups and across all ancestry groups. Inverse variance weighted fixed effects meta-analysis was performed with METAL (v. March25 2011)[76]. Heterogeneity between datasets was tested with a Cochran test and for nominally significant Q-values, a Han-Eskin random effects model (RE-HE) meta-analysis was performed with METASOFT v.2.0.1[77]. Markers with summary statistics in less than 25% of the total effective sample size or present in less than three studies were removed from meta-analyses. Quantile-quantile (QQ) plot of expected versus observed $-\log_{10}$ p-values included genotyped and imputed SNPs at MAF ≥1%. The proportion of inflation of test statistics due to the actual polygenic signal (rather than other causes such as population stratification) was estimated as 1—(LDSC intercept—1)/(mean observed Chi-square—1), using LD-score regression[12] (LDSC).

For primary analyses, genome-wide significance was declared at $P < 5 \times 10^{-8}$. To account for multiple comparisons in analyses stratified by sex, genome-wide significance was also considered at $P < 1.67 \times 10^{-8}$. For genome-wide significant hits, Forest plots and PM-Plots were generated using the programs METASOFT with default settings and M-values were generated using the MCMC option[13,78]. For a given study and SNP, the M-value is the posterior probability that there is a SNP effect in that study. Studies with values <0.1 are predicted to have no effect, values ≥0.1 and ≤0.9 are ambiguous, and values >0.9 are predicted to have an effect. In PM-plots, M-values are plotted against -$\log_{10}$ P-values. Regional association plots were generated using LocusZoom[79] with 400KB windows around the index variant and compared to the corresponding windows in the other ancestry groups, including the 1000 Genomes Nov. 2014 reference populations EUR, AFR and AMR, respectively. To test for sex-specific effects, a z-test was performed on the difference of the effect estimates from male and female sex-stratified analyses.

**Estimating PTSD heritability.** SNP-based heritability estimates ($h^2_{SNP}$) in EUA subjects were calculated using LDSC on meta-analysis summary data. Estimates were calculated for the combined PGC freeze 2 samples (PGC2) and separately for PGC1.5 (without UKB), the UK biobank (including alternative subject/phenotype selections), and for men and women. Unconstrained regression intercepts were used to account for potential inclusion of related subjects and residual population stratification, and precomputed LD scores from 1KGP EUR populations were used. For population prevalence we used a range of values (conservative low at 10%, moderate at 30%, and very high at 50%), based on prevalences reported for subjects exposed to different types of trauma[80]. Sample prevalence was set to the actual proportion of cases in each set of data.

To estimate $h^2_{SNP}$ in admixed individuals and compare $h^2_{SNP}$ across different ancestries, individual-level genotype data was analyzed using an unweighted linear mixed model[81] as implemented in the LDAK software[82]. For each ancestry group (EUA and AFA, respectively), imputed individual-level genotype data were filtered to bi-allelic SNPs with MAF ≥1% in the corresponding 1KGP phase 3 superpopulation. Imputed genotype probabilities ≥0.8 were converted to best-guess genotype calls, and for each ancestry group, studies were merged and SNPs with <95% genotyping rate or MAF <10% removed. Next, to estimate relatedness between subjects, a genetic relatedness matrix (GRM) was constructed based on autosomal SNPs that were LD pruned at $r^2 > 0.2$ over a 1MB window, and an unweighted model with α = −1, where α is the power parameter controlling the relationship between heritability and MAF. To prevent bias of $h^2_{SNP}$ due to cryptic relatedness, strict relatedness filters were applied. For pairs with relatedness values > the negative of the smallest observed kinship (−0.014 for EUA and −0.045 for AFA, respectively), one subject was randomly removed. PC's were then calculated in the remaining sets of unrelated subjects. Finally, to estimate $h^2_{SNP}$, an unweighted GRM was estimated without LD-pruning, and $h^2_{SNP}$ was calculated on the liability scale using REML in LDAK, including 5 PC's and dummy indicator variables for study (number of studies - 1) as covariates.

**Comparability of PGC2 studies.** To compare the genetic signal between specific PGC2 subsets, LDSC[12] was used to estimate heritability and genetic correlations. Small EUA studies with $N < 200$ cases and total effective sample size of $N < 500$ were selected ($N = 24$ studies; GWAS including 2102 cases and 7366 controls, effective $N = 5162$) and compared to larger studies. To reduce standard error given this relatively small sample, we estimated heritability with the LDSC intercept constrained to 1, after first testing that the intercept was not significantly different from 1.

**Replication study.** Data from the US Million Veteran Program (MVP) were used to replicate GWAS findings[9]. Participants reported here completed the PCL-C that asked respondents to report how much they have been bothered in the past 30 days by symptoms in response to stressful experiences (i.e. not just military experiences). The symptom cluster most distinctive for PTSD, re-experiencing symptoms (range 5–25), was analyzed. After accounting for missing phenotype data, the final sample for European Americans was 146,660, of whom 41.3% were combat-exposed.

Genotyping was accomplished via a 723,305-SNP Affymetrix Axiom biobank array, customized for the MVP. Imputation was performed with Minimac 3[83] and the 1000 Genomes Phase 3 reference panel. GWAS analysis was conducted using RVTEST[84] using linear regression with the first 10 principal components, age, and sex included as covariates. The results were filtered with imputation quality score $R^2 \geq 0.9$, MAF > 0.01 and HWE test $P$-value > $1 \times 10^{-06}$. LDSC was used to estimate genetic correlation with the PGC2 EUA sample. The PGC2 EUA GWAS summary statistics were used to estimate PRS in MVP samples, where linear regression was then used to test for association between PRS and re-experiencing symptoms.

**Local ancestry deconvolution.** A pipeline was developed to determine local ancestry in subjects with African and/or Native American admixture (AFA, AMA; Supplementary Fig. 28). Additional QC to consistently prepare cohort data for downstream analysis was performed with a custom script (https://github.com/eatkinson/Post-QC). Post-QC steps involved extracting autosomal data, removing duplicate loci, updating SNP IDs to dbSNP 144, orienting data to the 1KGP reference (with removal of indels and loci that either were not found in 1KGP or that had different coding alleles), flipping alleles that were on the wrong strand, and removing ambiguous SNPs.

Data harmonization and phasing: We then intersected and jointly phased the post-QC'ed cohort data with autosomal data from 247 1KGP reference panel individuals, removing conflicting sites and flipping any remaining strand flips. The merged dataset was then filtered to include only informative SNPs present in both the cohort and reference panel using a filter of MAF ≥ 0.05 and a genotype missingness cutoff of 90%. The program SHAPEIT2[85] was used to phase chromosomes, informed by the HapMap combined b37 recombination map[86]. Individuals from the cohort and reference panel were then separated and exported as harmonized sample and reference panel VCFs to be fed into RFMix[87].

Reference panel: Three ancestral populations of European, African, and Native American ancestry were chosen for the admixed AFA cohorts based on ancestry proportion estimates from SNPweights runs. All reference populations were taken from 1KGP phase 3 data[71]. Specifically, 108 West African Bantu-speaking YRI were used as the African reference population, 99 CEU comprised the European reference, and 40 PEL of >85% Native American ancestry were used as the Native American reference panel. Individuals used as the reference panel can be found on (https://github.com/eatkinson).

Local ancestry inference (LAI) parameters: LAI was run on each cohort separately using RFMix version 2[87] (https://github.com/slowkoni/rfmix) with 1 EM iteration and a window size of 0.2 cM. We used the HapMap b37 recombination map[86] to inform switches. The -n 5 flag (terminal node size for random forest trees) was included to account for an unequal number of reference individuals per reference population. We additionally used the --reanalyze-reference flag, which recalculates admixture in the reference samples for improved ability to distinguish ancestries.

Local ancestry of genome-wide significant variants: Haplotypes of the genomic regions around genome-wide significant associations were aligned to the local ancestry calls according to genomic position. To compare MAF of top hits on different ancestral backgrounds within a specific admixed population (AFA or AMA), subjects were grouped according to the number of copies (1 or 2) of a specific ancestry (European, African, and Native American, respectively) at that position. For a given SNP, MAF was calculated within each of the six groups. To ensure successful elimination of population stratification by standard global PC's in regression analyses of admixed populations, two (out of 3, to reduce redundancy) local ancestry dosage covariates were included, coded as the number of copies (0, 1 or 2) from a given ancestral background. Finally, to compare if effects of the minor allele depend on a specific ancestral background (European, African, and Native American), for each SNP, we coded variables that counted the number of copies of the minor allele per ancestral background. Association between these three variables and PTSD were jointly evaluated using a logistic regression, including study indicators and five global ancestry PC's as additional covariates.

**Functional mapping and annotation.** We used Functional Mapping and Annotation of genetic associations (FUMA) v1.3.0 (https://fuma.ctglab.nl/) to annotate GWAS data and obtain functional characterization of risk loci. Annotations are based on human genome assembly GRCh37 (hg19). FUMA was used with default settings unless stated otherwise. The SNP2Gene module was used to define independent genomic risk loci and variants in LD with lead SNPs ($r^2 > 0.6$, calculated using ancestry appropriate 1KGP reference genotypes). SNPs in risk loci were mapped to protein-coding genes with a 10 kb window. Functional consequences for SNPs were obtained by mapping the SNPs on their chromosomal position and reference alleles to databases containing known functional annotations, including ANNOVAR, Combined Annotation Dependent Depletion (CADD), RegulomeDB (RDB), and chromatin states (only brain tissues/cell types were selected). Next eQTL mapping was performed on significant (FDR $q < 0.05$) SNP-gene pairs, mapping to GTEx v7 brain tissue, RNA-seq data from the CommonMind Consortium and the BRAINEAC database. Chromatin interaction mapping was performed using built-in chromatin interaction data from the dorsolateral prefrontal cortex, hippocampus and neuronal progenitor cell line. We used a FDR $q < 1 \times 10^{-5}$ to define significant interactions, based on previous recommendations,

modified to account for the differences in cell lines used here. SNPs were also checked for previously reported phenotypic associations in published GWAS listed in the NHGRI-EBI catalog.

**Gene-based and gene-set analysis with MAGMA**. Gene-based analysis was performed with the FUMA implementation of MAGMA. SNPs were mapped to 18,222 protein-coding genes. For each gene, its association with PTSD was determined as the weighted mean $\chi^2$ test statistic of SNPs mapped to the gene, where LD patterns were calculated using ancestry appropriate 1KGP reference genotypes. Significance of genes was set at a Bonferroni-corrected threshold of $P = 0.05/18,222 = 2.7 \times 10^{-6}$.

To see if specific biological pathways were implicated in PTSD, gene-based test statistics were used to perform a competitive set-based analysis of 10,894 pre-defined curated gene sets and GO terms obtained from MsigDB using MAGMA. Significance of pathways was set at a Bonferroni-corrected threshold of $P = 0.05/10,894 = 4.6 \times 10^{-6}$. To test if tissue-specific gene expression was associated with PTSD, gene-set-based analysis was also used with expression data from GTEx v7 RNA-seq and BrainSpan RNA-seq, where the expression of genes within specific tissues were used to define the gene properties used in the gene-set analysis model.

**Functional follow-up of the AFA top hit rs115539978**. *Cell Culture Experiments, RNA extraction and qPCR*: Lymphoblastoid cell lines (LCLs) from the AFR superpopulation were obtained from the Coriell Institute, NJ (Supplementary Table 13, $N = 6$ lines each for the homozygous major and homozygous minor allele). Cells were cultured in RPMI 1640 medium with GlutaMAX (Thermo Scientific, 61870-036) supplemented with 15% FBS (Thermo Scientific, 26140079) and 1X Antibiotic-Antimycotic (Thermo Scientific, 15240-062) at 37 C and 5% CO2 in a humidified incubator. For Dexamethasone (Dex) treatment, a final concentration of 100 nM Dex (Sigma–Aldrich) in 100% Ethanol was added to the medium for a total of 4 hr. All experiments were run in duplicates.

RNA was extracted using the Quick-RNA MiniPrep Kit (Zymo, R2060) according to the instructions of the manufacturer including an additional DNase digestion. RNA concentrations were quantified via Qubit and cDNA was generated using the SuperScript IV First Strand Kit (Life Technologies, 18091200) according to the manufacturer's instructions. SYBR green qPCR reactions were carried out in duplicates using POWERUP SYBR Green Master Mix (Life Technologies, A25743) and custom primer pairs (Supplementary Table 14) according to the manufacturer's recommendations. Data were analyzed using the ΔΔCt method[88] and GAPDH as reference. Between group differences were calculated using one-way ANCOVA with sex as covariate. Significance threshold was set at $P = 0.05$.

**Deep phenotyping of the AFA top hit rs115539978**. *Neuroimaging*: Scanning of 87 GTPC subjects took place on a 3.0 T Siemens Trio with echo-planar imaging (Siemens, Malvern, PA). High-resolution T1-weighted anatomical scans were collected using a 3D MP-RAGE sequence, with 176 contiguous 1 mm sagittal slices (TR/TE/TI = 2000/3.02/900 ms, 1 mm$^3$ voxel size). T1 images were processed in Freesurfer version 5.3 (https://surfer.nmr.mgh.harvard.edu). Gray matter volume from subcortical structures was extracted through automated segmentation, and data quality checks were performed following the ENIGMA 2 protocol (http://enigma.ini.usc.edu/protocols/imaging-protocols/), a method designed to standardize quality control procedures across laboratories to facilitate replication. Briefly, segmented T1 images were visually examined for errors, and summary statistics and a summary of outliers ± 3 SD from the mean were generated from the segmentation of the left and right amygdala and hippocampus. Regional volumes that were visually confirmed to contain a segmentation error were discarded.

*Startle Physiology*: The physiological data of 299 GTPC subjects were acquired using Biopac MP150 for Windows (Biopac Systems, Inc., Aero Camino, CA). The acquired data were filtered, rectified, and smoothed using MindWare software (MindWare Technologies, Ltd., Gahanna, OH) and exported for statistical analyses. Startle data were collected by recording the eyeblink muscle contraction using the electromyography (EMG) module of the Biopac system. The startle response was recorded with two Ag/AgCl electrodes; one was placed on the orbicularis oculi muscle below the pupil and the other 1 cm lateral to the first electrode. A common ground electrode was placed on the palm. Impedance levels were less than 6 kilo-ohms for each participant. The startle probe was a 108-dB(A) SPL, 40 ms burst of broadband noise delivered through headphones (Maico, TDH-39-P). The maximum amplitude of the eyeblink muscle contraction 20–200 ms after presentation of the startle probe was used as a measure of startle magnitude.

**Polygenic scoring**. Polygenic risk scores (PRS) were calculated in hold out target samples based on SNP effect sizes from PTSD GWAS in non-overlapping discovery/training samples. GWAS summary statistics were filtered to common (MAF > 5%), well imputed variants (INFO > 0.9). Indels, ambiguous SNPs, and variants in the extended MHC region (chr6:25-34 Mb) were removed. LD pruning was performed using the --clump procedure in PLINK1.9, where variants were pruned if they were nearby (within 500 kb) and in LD ($r^2 > 0.3$) with the leading variant (lowest $P$-value) in a given region. PRS were calculated in PRSice v2.1.2 using the best-guess genotype data of target samples, where for each SNP the risk score was estimated as the natural log of the odds ratio multiplied by number of

copies of the risk allele. PRS was estimated as the sum of risk scores overall SNPs. PRS were generated at multiple $P$-value thresholds ($P_T$) (at intervals of 0.01 ranging from $P = 0.0001$ to $P = 1$). Best-fit PRS (at $P_T = 0.3$ for PTSD and $P_T = 0.3$ for re-experiencing symptoms, respectively) were used to predict PTSD status under logistic regression, adjusting for 5 PCs and dummy study indicator variables, using the glm function in R 3.2.1. PRS prediction plots were based on quintiles of PRS, with odds ratios calculated in reference to the lowest quintile. The proportion of variance explained by PRS was estimated as the difference in Nagelkerke's $R^2$ between a model including PRS plus covariates and a model with only covariates. $R^2$ was converted to the liability scale assuming a 30% prevalence, using the equation found in Lee et al[89]. $P$-values for PRS were derived from a likelihood ratio test comparing the two models.

**Genetic correlation of PTSD with other traits and disorders**. Bivariate LD-score regression (LDSC) was used to calculate pairwise genetic correlation ($r_g$) between PTSD and 235 traits with publicly available GWAS summary statistics on LD Hub[12]. Summary statistics for PTSD studies were restricted to the EUA meta-analysis, including UKB subjects (23,212 cases, 151,447 controls) and significance was evaluated based on a conservative Bonferroni correction for 235 phenotypes (i.e. correlated traits and traits measured twice in independent studies were counted independently).

In addition, these phenotypes were compared with genetic correlations reported for PTSD and several psychiatric disorders, including 221 phenotypes and MDD[18], 172 phenotypes and Schizophrenia (SCZ)[10], 196 phenotypes and bipolar disorder (BPD)[19] and 219 phenotypes and attention-deficit/hyperactivity disorder (ADHD)[20]. Due to substantial overlap with other traits, two education, four anthropometric and two cancer phenotypes were omitted.

**Conditional analyses to test for disease specific effects**. To evaluate if the effects of the top variants identified in the PTSD GWAS were specific to PTSD, we conditioned PTSD on MDD, and MDD plus BPD plus SCZ using the multi-trait conditional and joint analysis (mtCOJO)[24] feature in GCTA to regress out the effects of correlated traits based on external GWAS summary data. MDD was selected here as the main psychiatric trait because of the high co-morbidity and genetic correlation of depressive symptoms and PTSD ($r_g = 0.80$ for depressive symptoms and $r_g = 0.62$ for MDD; see Supplementary Data 3). Publicly available summary statistics were supplied as program inputs: Bipolar cases vs. controls for BPD, and MDD2 excluding 23andMe for MDD (both from https://www.med. unc. edu/pgc/results-and-downloads); Schizophrenia: CLOZUK + PGC2 meta-analysis for SCZ (http://walters.psycm.cf.ac.uk/). The effect of each psychiatric disorder on PTSD was estimated using a generalized summary-data based Mendelian randomization analysis of significant LD independent psychiatric trait SNPs ($r^2 < 0.05$, based on 1000 G Phase 3 CEU samples), where the threshold for significance was set to $P < 5 \times 10^{-7}$ due to having less than the required 10 significant independent SNPs at the program default $P < 5 \times 10^{-8}$ for MDD. Estimates of heritability, genetic correlation, and sample overlap of psychiatric trait and PTSD GWAS were estimated using precomputed LD scores based on 1000 G Europeans that were supplied with LDSC (https://data.broadinstitute.org/alkesgroup/LDSCORE/eur_w_ld_chr.tar.bz2).

**Reporting summary**. Further information on research design is available in the Nature Research Reporting Summary linked to this article.

## Data availability
The full meta-analyses summary statistics are available for download from the Psychiatric Genomics Consortium at https://www.med.unc.edu/pgc/results-and-downloads/. Access to individual-level data for available datasets may be requested through the PGC Data Access Portal at https://www.med.unc.edu/pgc/shared-methods/data-access-portal/. All other data that support the findings of this study are available from the corresponding authors upon request.

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

## Acknowledgements

This work was funded by Cohen Veterans Bioscience, the NIMH/U.S. Army Medical Research and Materiel Command Grant R01MH106595 to C.M.N., I.L., K.J.R. and K.C.K., One Mind, and supported by 5U01MH109539 to the Psychiatric Genomics Consortium. Statistical Analysis were carried out on the NL Genetic Cluster computer (URL) hosted by SURFsara. Genotyping of samples was supported in part through the Stanley Center for Psychiatric Genetics at the Broad Institute of MIT and Harvard. This research has been conducted using the UK biobank resource under application number 16577. This work would not have been possible without the contributions of the investigators who comprise the PGC-PTSD working group, and especially the more than 206,000 research participants worldwide who shared their life experiences and biological samples with PGC-PTSD investigators. Full acknowledgements are in the Supplementary Note 2.

## Author contributions

PGC-PTSD management group: M.H., K.C.K., I.L., C.M.N., A.C.P., K.J.R. Writing group: E.G.A., C.-Y.C., K.W.C., J.R.I.C., S.D., L.E.D., J.G., T.K., K.C.K., D.F.L., M.W.L., A.X.M., C.M.N., R.P., A. Ratanatharathorn, K.J.R., M.B.S., K.T. Study PI or co-PI: A.E.A., A.B.A., S.B. Andersen, P.A.A., S.B. Austin, E.A., D.B., D.G.B., J.C.B., L.J.B., J.I.B., A.D.B., B.B., G.B., J.R.C., M.J.D., J.D., D.L.D., K.D., A.D.-K., C.R.E., C.E.F., L.A.F., N.C.F., B.G., J.G., E.G., C.G., A.G.U., M.A.H., A.C.H., D.M.H., M. Jakovljevic, I.J., T.J., K.-I.K., M.L.K., R.C.K., N.A.K., K.C.K., H.R.K., W.S.K., B.R.L., I.L., M.J.L., C.M., N.G.M., M.R.M., R.E.M., K.A.M., S.A.M., D. Mehta, W.P.M., M.W.M., C.P.M., O.M., P.B.M, B.M.N., E.C.N., C.M.N., M.N., S.B.N., A.L.P., R.H.P., M.A.P., K.J.R., V.B.R., P.R.B., K.R., S.E.S., S.S., J.S.S., A.K.S., J.W.S., S.R.S., D.J.S., M.B.S., R.J.U., E.V., J.V., Z.W., T.W., D.E.W., C.W., R.Y., R.M.Y, H.Z., L.A.Z. Obtained funding for studies: A.B.A., P.A.A., S.B. Austin, J.C.B., L.J.B., A.D.B., B.B., G.B., J.D., C.R.E., N.C.F., J.D.F., C.E.F., E.G., C.G., M.H., R.H., M.A.H., A.C.H., D.M.H., M. Jett, E.O.J., T.J., K.-I.K., N.A.K., K.C.K., W.S.K., B.R.L., I.L., M.J.L., C.M., N.G.M., R.E.M., K.A.M., S.A.M., W.P.M., M.W.M., C.P.M., O.M., P.B.M, E.C.N., C.M.N., M.N., M.A.P., K.J.R., B.O.R., A.K.S., S.R.S., M.H.T., R.J.U., E.V., J.V., Z.W., T.W., D.E.W., R.Y., R.M.Y, L.A.Z. Clinical: P.A.A., E.A., D.B., D.G.B., J.C.B., L.J.B., E.A.B., A.D.B., M.B., A.C.B., J.R.C., M.F.D., S.G.D., A.D.-K., C.R.E., N.C.F., J.D.F., C.E.F., S.G., E.G., A.G.U., G.G., R.H., D.M.H., M. Jakovljevic, E.O.J., A.G.J., K.-I.K., M.L.K., A.K., N.A.K., N.K., W.S.K., B.R.L., L.A.M.L., C.E.L., M.J.L., J.M.-K., D. Maurer, S.A.M., S.M., P.B.M, H.K.O., M.S.P., E.S.P., A.L.P., M.P., M.A.P., A.O.R., B.O.R., A.V.S., J.S.S., C.M.S., J.A.S., M.H.T., W.K.T., E.T., M.U., L.L.V.D.H., E.V., Y.W., Z.W., T.W., M.A.W., D.E.W., S.W., E.J.W., J.D.W., K.A.Y., R.Y., L.A.Z. Contributed data: O.A.A., P.A.A., D.G.B., L.J.B., A.D.B., M.B., R.A.B., J.R.C., J.M.C.-D.-A., A.M.D., D.L.D., C.R.E., A.E., N.C.F., D.F., C.E.F., S.G., B.G., S.M.J.H., D.M.H., I.J., A.G.J., M.L.K., R.C.K., A.P.K., W.S.K., L.A.M.L., C.E.L., I.L., B.L., M.J.L., M.R.M., A.M., K.A.M., O.M., P.B.M, C.M.N., M.N., S.B.N., M.O., M.S.P., E.S.P., A.L.P., M.P., M.A.P., K.J.R., V.B.R., P.R.B., A. Rung, S.E.S., S.S., J.S.S., D. Silove, S.R.S., M.B.S., E.T., L.L.V.D.H., M.V.H., T.W., D.E.W., S.W., J.D.W., R.Y., K.A.Y., L.A.Z. Statistical analysis: L.M.A., A.E.A.-K., E.G.A., C.-Y.C., J.R.I.C., S.G.D., L.E.D., M.E.G., B.G., S.D.G., G.G., X.-J.Q., D.F.L., M.W.L., A.L, A.X.M., C.M., A.R.M., S.M., D. Mehta, R.A.M., C.M.N., M.P., R.P., J.P.R., S.R., A.L.R., N.L.S., D. Schijven, C.M.S. Bioinformatics: L.M.A., A.E.A.-K., E.G.A., M.P.B., C.-Y.C., J.R.I.C., N.P.D, S.G.D., M.E.G., G.G., S.D.L., A.L, A.X.M., A.R.M., D. Mehta, D. Schijven, C.W. Genomics: M.B.-H., M.P.B., J.B.-G., K.D., M.H., S.H., M.A.H., T.K., S.D.L., J.J.L., A.C.P., K.J.R., B.P.F.R., D. Schijven, C.H.V., D.E.W.

## Competing interests

L.J.B., J.P.R., and the spouse of N.L.S. are listed as inventors on Issued U.S. Patent 8,080,371, "Markers for Addiction," covering the use of certain SNPs in determining the diagnosis, prognosis, and treatment of addiction. A.M.D. is a Founder of and holds equity in CorTechs Labs, Inc, and serves on its Scientific Advisory Board. He is a member of the Scientific Advisory Board of Human Longevity, Inc. and receives funding through research agreements with General Electric Healthcare and Medtronic, Inc. The terms of these arrangements have been reviewed and approved by UCSD in accordance with its conflict of interest policies. M.H. and A.C.P. are both employees of CVB, a Sponsor (non-profit) of the study. In the past 3 years, R.C.K. received support for his epidemiological studies from Sanofi Aventis; was a consultant for Johnson & Johnson Wellness and Prevention, Sage Pharmaceuticals, Shire, Takeda; and served on an advisory board for the Johnson & Johnson Services Inc. Lake Nona Life Project. Kessler is a co-owner of DataStat, Inc., a market research firm that carries out healthcare research. H.R.K. is a member of the American Society of Clinical Psychopharmacology's Alcohol Clinical Trials Initiative (ACTIVE), which in the last three years was supported by AbbVie, Alkermes, Amygdala Neurosciences, Arbor, Ethypharm, Indivior, Lilly, Lundbeck, Otsuka, and Pfizer. He is also named as an inventor on PCT patent application #15/878,640 entitled: "Genotype-guided dosing of opioid agonists," filed January 24, 2018. B. M.N. is a member, Scientific Advisory Board of Deep Genomics, a consultant for Camp4 Therapeutics Corporation, Merck & Co. and Avanir Pharmaceuticals, Inc. B.O.R. owns equity in Virtually Better, Inc. that creates virtual reality products. The terms of this arrangement have been reviewed and approved by Emory University in accordance with its conflict of interest policies. In the past 3 years, D.J.S. has received research grants and/or consultancy honoraria from Biocodex, Lundbeck, Servier, and Sun. M.B.S. has in the past three years been a consultant for Actelion, Aptinyx, Bionomics, Dart Neuroscience, Healthcare Management Technologies, Janssen, Neurocrine Biosciences, Oxeia Biopharmaceuticals, Pfizer, and Resilience Therapeutics. R.Y. is a co-inventor of the following patent application: "Genes associated with post-traumatic-stress disorder. European Patent# EP 2334816 B1". T.W. has acted as scientific advisor to H. Lundbeck A/S. All remaining authors declare no competing interests.

## Additional information

Caroline M. Nievergelt[1,2,3*], Adam X. Maihofer[1,2,3], Torsten Klengel[4,5,6], Elizabeth G. Atkinson[7,8], Chia-Yen Chen[7,8,9], Karmel W. Choi[7,10,11], Jonathan R.I. Coleman[12,13], Shareefa Dalvie[14], Laramie E. Duncan[15], Joel Gelernter[16,17,18], Daniel F. Levey[18,19], Mark W. Logue[20], Renato Polimanti[18,19], Allison C. Provost[21], Andrew Ratanatharathorn[11], Murray B. Stein[1,22,23], Katy Torres[1,2,3], Allison E. Aiello[24], Lynn M. Almli[25], Ananda B. Amstadter[26], Søren B. Andersen[27], Ole A. Andreassen[28], Paul A. Arbisi[29], Allison E. Ashley-Koch[30], S. Bryn Austin[4,31,32,33], Esmina Avdibegovic[34], Dragan Babić[35], Marie Bækvad-Hansen[36,37], Dewleen G. Baker[1,2,23], Jean C. Beckham[38,39,40], Laura J. Bierut[41], Jonathan I. Bisson[42], Marco P. Boks[43], Elizabeth A. Bolger[4,5], Anders D. Børglum[37,44,45], Bekh Bradley[46,25], Megan Brashear[47], Gerome Breen[12,13], Richard A. Bryant[48], Angela C. Bustamante[49], Jonas Bybjerg-Grauholm[36,37], Joseph R. Calabrese[50], José M. Caldas- de- Almeida[51], Anders M. Dale[52], Mark J. Daly[9], Nikolaos P. Daskalakis[4,5,21,53], Jürgen Deckert[54], Douglas L. Delahanty[55,56], Michelle F. Dennis[39,38,40], Seth G. Disner[57], Katharina Domschke[58,59], Alma Dzubur-Kulenovic[60], Christopher R. Erbes[61,62], Alexandra Evans[63], Lindsay A. Farrer[64], Norah C. Feeny[65], Janine D. Flory[53], David Forbes[66], Carol E. Franz[1], Sandro Galea[67], Melanie E. Garrett[39], Bizu Gelaye[11], Elbert Geuze[68,69], Charles Gillespie[25], Aferdita Goci Uka[70], Scott D. Gordon[71], Guia Guffanti[4,5], Rasha Hammamieh[72], Supriya Harnal[7], Michael A. Hauser[39], Andrew C. Heath[73], Sian M.J. Hemmings[74], David Michael Hougaard[36,37], Miro Jakovljevic[75], Marti Jett[72], Eric Otto Johnson[76], Ian Jones[63], Tanja Jovanovic[25], Xue-Jun Qin[30], Angela G. Junglen[55], Karen-Inge Karstoft[27,77], Milissa L. Kaufman[4,5], Ronald C. Kessler[4], Alaptagin Khan[78,5], Nathan A. Kimbrel[30,38,40], Anthony P. King[79], Nastassja Koen[14], Henry R. Kranzler[80,81], William S. Kremen[1,2], Bruce R. Lawford[82,83], Lauren A.M. Lebois[4,5], Catrin E. Lewis[63], Sarah D. Linnstaedt[84], Adriana Lori[85], Bozo Lugonja[63], Jurjen J. Luykx[43,69], Michael J. Lyons[86], Jessica Maples-Keller[25], Charles Marmar[87], Alicia R. Martin[8,7], Nicholas G. Martin[71], Douglas Maurer[88], Matig R. Mavissakalian[50], Alexander McFarlane[89], Regina E. McGlinchey[90], Katie A. McLaughlin[91], Samuel A. McLean[84,92], Sarah McLeay[93], Divya Mehta[82,94], William P. Milberg[90], Mark W. Miller[20], Rajendra A. Morey[30], Charles Phillip Morris[82,83], Ole Mors[95,37], Preben B. Mortensen[96,44,97,37], Benjamin M. Neale[8,7], Elliot C. Nelson[41], Merete Nordentoft[37,98], Sonya B. Norman[1,99,100], Meaghan O'Donnell[66], Holly K. Orcutt[101], Matthew S. Panizzon[1], Edward S. Peters[47], Alan L. Peterson[102], Matthew Peverill[103], Robert H. Pietrzak[19,104], Melissa A. Polusny[105,106,61], John P. Rice[41], Stephan Ripke[7,9,107], Victoria B. Risbrough[2,1,3], Andrea L. Roberts[108], Alex O. Rothbaum[65], Barbara O. Rothbaum[25], Peter Roy-Byrne[103], Ken Ruggiero[109], Ariane Rung[47], Bart P.F. Rutten[110], Nancy L. Saccone[41], Sixto E. Sanchez[111], Dick Schijven[43,69], Soraya Seedat[74], Antonia V. Seligowski[4,5], Julia S. Seng[112], Christina M. Sheerin[26], Derrick Silove[113], Alicia K. Smith[25,85], Jordan W. Smoller[8,10,7],

Scott R. Sponheim [29,61], Dan J. Stein [14], Jennifer S. Stevens [25], Jennifer A. Sumner[114], Martin H. Teicher[4,5], Wesley K. Thompson[1,37,115,116], Edward Trapido[47], Monica Uddin[117], Robert J. Ursano[118], Leigh Luella van den Heuvel[74], Miranda Van Hooff[89], Eric Vermetten[87,119,120,121], Christiaan H. Vinkers [122,123], Joanne Voisey [82,83], Yunpeng Wang [37,115,116], Zhewu Wang[124,125], Thomas Werge [37,115,126], Michelle A. Williams [11], Douglas E. Williamson[39,38], Sherry Winternitz[4,5], Christiane Wolf[54], Erika J. Wolf[20], Jonathan D. Wolff[5], Rachel Yehuda[53,127], Ross McD. Young[82,94], Keith A. Young [128,129], Hongyu Zhao [130], Lori A. Zoellner[131], Israel Liberzon[79], Kerry J. Ressler[4,5,25], Magali Haas [21] & Karestan C. Koenen[7,9,132]

[1]University of California San Diego, Department of Psychiatry, La Jolla, CA, USA. [2]Veterans Affairs San Diego Healthcare System, Center of Excellence for Stress and Mental Health, San Diego, CA, USA. [3]Veterans Affairs San Diego Healthcare System, Research Service, San Diego, CA, USA. [4]Harvard Medical School, Department of Psychiatry, Boston, MA, USA. [5]McLean Hospital, Belmont, MA, USA. [6]University Medical Center Goettingen, Department of Psychiatry, Göttingen, DE, Germany. [7]Broad Institute of MIT and Harvard, Stanley Center for Psychiatric Research, Cambridge, MA, USA. [8]Massachusetts General Hospital, Analytic and Translational Genetics Unit, Boston, MA, USA. [9]Massachusetts General Hospital, Psychiatric and Neurodevelopmental Genetics Unit (PNGU), Boston, MA, USA. [10]Massachusetts General Hospital, Department of Psychiatry, Boston, MA, USA. [11]Harvard T.H. Chan School of Public Health, Department of Epidemiology, Boston, MA, USA. [12]King's College London, Social, Genetic and Developmental Psychiatry Centre, Institute of Psychiatry, Psychology and Neuroscience, London, GB, USA. [13]King's College London, NIHR BRC at the Maudsley, London, GB, USA. [14]University of Cape Town, SA MRC Unit on Risk & Resilience in Mental Disorders, Department of Psychiatry, Cape Town, Western Cape, ZA, USA. [15]Stanford University, Department of Psychiatry and Behavioral Sciences, Stanford, CA, USA. [16]US Department of Veterans Affairs, Department of Psychiatry, West Haven, CT, USA. [17]Yale University School of Medicine, Department of Genetics and Neuroscience, New Haven, CT, USA. [18]VA Connecticut Healthcare Center, West Haven, CT, USA. [19]Yale University School of Medicine, Department of Psychiatry, New Haven, CT, USA. [20]VA Boston Healthcare System, National Center for PTSD, Boston, MA, USA. [21]Cohen Veterans Bioscience, Cambridge, MA, USA. [22]Veterans Affairs San Diego Healthcare System, Million Veteran Program, San Diego, CA, USA. [23]Veterans Affairs San Diego Healthcare System, Psychiatry Service, San Diego, CA, USA. [24]Gillings School of Global Public Health, Department of Epidemiology, Chapel Hill, NC, USA. [25]Emory University, Department of Psychiatry and Behavioral Sciences, Atlanta, GA, USA. [26]Virginia Institute for Psychiatric and Behavioral Genetics, Department of Psychiatry, Richmond, VA, USA. [27]The Danish Veteran Centre, Research and Knowledge Centre, Ringsted, Sjaelland, Denmark. [28]University of Oslo, Institute of Clinical Medicine, Oslo, NO, Norway. [29]Minneapolis VA Health Care System, Mental Health Service Line, Minneapolis, MN, USA. [30]Duke University, Duke Molecular Physiology Institute, Durham, NC, USA. [31]Boston Children's Hospital, Division of Adolescent and Young Adult Medicine, Boston, MA, USA. [32]Brigham and Women's Hospital, Channing Division of Network Medicine, Boston, MA, USA. [33]Harvard School of Public Health, Department of Social and Behavioral Sciences, Boston, MA, USA. [34]University Clinical Center of Tuzla, Department of Psychiatry, Tuzla, BA, Bosnia and Herzegovina. [35]University Clinical Center of Mostar, Department of Psychiatry, Mostar, BA, Bosnia and Herzegovina. [36]Statens Serum Institut, Department for Congenital Disorders, Copenhagen, DK, Denmark. [37]The Lundbeck Foundation Initiative for Integrative Psychiatric Research, iPSYCH, DK, Denmark. [38]Durham VA Medical Center, Research, Durham, NC, USA. [39]Duke University, Department of Psychiatry and Behavioral Sciences, Durham, NC, USA. [40]VA Mid-Atlantic Mental Illness Research, Education, and Clinical Center (MIRECC), Genetics Research Laboratory, Durham, NC, USA. [41]Washington University in Saint Louis School of Medicine, Department of Psychiatry, Saint Louis, MO, USA. [42]Cardiff University, National Centre for Mental Health, MRC Centre for Psychiatric Genetics and Genomics, Cardiff, UK. [43]UMC Utrecht Brain Center Rudolf Magnus, Department of Translational Neuroscience, Utrecht, Utrecht, NL, Netherlands. [44]Aarhus University, Centre for Integrative Sequencing, iSEQ, Aarhus, DK, Denmark. [45]Aarhus University, Department of Biomedicine - Human Genetics, Aarhus, DK, Denmark. [46]Atlanta VA Health Care System, Mental Health Service Line, Decatur, GA, USA. [47]Louisiana State University Health Sciences Center, School of Public Health and Department of Epidemiology, New Orleans, LA, USA. [48]University of New South Wales, Department of Psychology, Sydney, NSW, Australia. [49]University of Michigan Medical School, Division of Pulmonary and Critical Care Medicine, Department of Internal Medicine, Ann Arbor, MI, USA. [50]University Hospitals, Department of Psychiatry, Cleveland, OH, USA. [51]CEDOC -Chronic Diseases Research Centre, Lisbon Institute of Global Mental Health, Lisbon, PT, Portugal. [52]University of California San Diego, Department of Radiology, Department of Neurosciences, La Jolla, CA, USA. [53]Icahn School of Medicine at Mount Sinai, Department of Psychiatry, New York, NY, USA. [54]University Hospital of Würzburg, Center of Mental Health, Psychiatry, Psychosomatics and Psychotherapy, Würzburg, DE, Germany. [55]Kent State University, Department of Psychological Sciences, Kent, OH, USA. [56]Kent State University, Research and Sponsored Programs, Kent, OH, USA. [57]Minneapolis VA Health Care System, Research Service Line, Minneapolis, MN, USA. [58]Medical Center-University of Freiburg, Faculty of Medicine, Department of Psychiatry and Psychotherapy, Freiburg, DE, Germany. [59]University of Freiburg, Faculty of Medicine, Centre for Basics in Neuromodulation, Freiburg, DE, Germany. [60]University Clinical Center of Sarajevo, Department of Psychiatry, Sarajevo, BA, Bosnia and Herzegovina. [61]University of Minnesota, Department of Psychiatry, Minneapolis, MN, USA. [62]Minneapolis VA Health Care System, Center for Care Delivery and Outcomes Research (CCDOR), Minneapolis, MN, USA. [63]Cardiff University, National Centre for Mental Health, MRC Centre for Psychiatric Genetics and Genomics, Cardiff, South Glamorgan, GB, USA. [64]Boston University School of Medicine, Department of Medicine, Boston, MA, USA. [65]Case Western Reserve University, Department of Psychological Sciences, Cleveland, OH, USA. [66]University of Melbourne, Department of Psychiatry, Melbourne, VIC, AU, USA. [67]Boston University, Department of Psychological and Brain Sciences, Boston, MA, USA. [68]Netherlands Ministry of Defence, Brain Research and Innovation Centre, Utrecht, Utrecht, NL, Netherlands. [69]UMC Utrecht Brain Center Rudolf Magnus, Department of Psychiatry, Utrecht, Utrecht, NL, Netherlands. [70]University Clinical Centre of Kosovo, Department of Psychiatry, Prishtina, Kosovo, XK, USA. [71]QIMR Berghofer Medical Research Institute, Department of Genetics and Computational Biology, Brisbane, Queensland, Australia. [72]US Army Medical Research and Materiel Command, USACEHR, Fort Detrick, MD, USA. [73]Washington University in Saint Louis School of Medicine, Department of Genetics, Saint Louis, MO, USA. [74]Stellenbosch University Faculty of Medicine and Health Sciences, Department of Psychiatry, Cape Town, Western Cape, ZA, South Africa. [75]University Hospital Center of Zagreb, Department of Psychiatry, Zagreb, HR, USA. [76]RTI International, Behavioral Health and Criminal Justice Division, Research Triangle Park, NC, USA. [77]University of Copenhagen, Department of Psychology, Copenhagen, DK, Denmark. [78]Harvard Medical School, Department of Health Care Policy, Boston, MA, USA. [79]University of Michigan Medical School, Department of Psychiatry, Ann Arbor, MI, USA. [80]University of Pennsylvania Perelman School of Medicine, Department of Psychiatry, Philadelphia, PA, USA. [81]Mental Illness Research, Education and Clinical Center, Crescenz VAMC, Philadelphia, PA, USA. [82]Queensland University of Technology, Institute of Health and Biomedical Innovation, Kelvin Grove, QLD, AU, Australia. [83]Queensland University of Technology, School of Biomedical Sciences, Kelvin Grove, QLD, AU, Australia. [84]UNC Institute for Trauma Recovery, Department of Anesthesiology, Chapel Hill, NC, USA.

[85]Emory University, Department of Gynecology and Obstetrics, Atlanta, GA, USA. [86]Boston University, Dean's Office, Boston, MA, USA. [87]New York University School of Medicine, Department of Psychiatry, New York, NY, USA. [88]United States Army, Command, Fort Sill, OK, USA. [89]University of Adelaide, Department of Psychiatry, Adelaide, South Australia, AU, Australia. [90]VA Boston Health Care System, GRECC/TRACTS, Boston, MA, USA. [91]Harvard University, Department of Psychology, Boston, MA, USA. [92]UNC Institute for Trauma Recovery, Department of Emergency Medicine, Chapel Hill, NC, USA. [93]Gallipoli Medical Research Institute, PTSD Initiative, Greenslopes, Queensland, AU, Australia. [94]Queensland University of Technology, School of Psychology and Counseling, Faculty of Health, Kelvin Grove, QLD, AU, Australia. [95]Aarhus University Hospital, Psychosis Research Unit, Risskov, DK, Denmark. [96]Aarhus University, Centre for Integrated Register-based Research, Aarhus, DK, Denmark. [97]Aarhus University, National Centre for Register-Based Research, Aarhus, DK, Denmark. [98]University of Copenhagen, Mental Health Services in the Capital Region of Denmark, Mental Health Center Copenhagen, Copenhagen, DK, Denmark. [99]Veterans Affairs San Diego Healthcare System, Department of Research and Psychiatry, San Diego, CA, USA. [100]National Center for Post Traumatic Stress Disorder, Executive Division, White River Junction, San Diego, VT, USA. [101]Northern Illinois University, Department of Psychology, DeKalb, IL, USA. [102]University of Texas Health Science Center at San Antonio, Department of Psychiatry, San Antonio, TX, USA. [103]University of Washington, Department of Psychology, Seattle, WA, USA. [104]U.S. Department of Veterans Affairs National Center for Posttraumatic Stress Disorder, West Haven, CT, USA. [105]Minneapolis VA Health Care System, Department of Mental Health, Minneapolis, MN, USA. [106]Minneapolis VA Health Care System, Department of Psychology, Minneapolis, MN, USA. [107]Charité - Universitätsmedizin, Department of Psychiatry and Psychotherapy, Berlin, GE, Germany. [108]Harvard T.H. Chan School of Public Health, Department of Environmental Health, Boston, MA, USA. [109]Medical University of South Carolina, Department of Nursing and Department of Psychiatry, Charleston, SC, USA. [110]Maastricht Universitair Medisch Centrum, School for Mental Health and Neuroscience, Department of Psychiatry and Neuropsychology, Maastricht, Limburg, NL, Netherlands. [111]Universidad Peruana de Ciencias Aplicadas Facultad de Ciencias de la Salud, Department of Medicine, Lima, Lima, PE, USA. [112]University of Michigan, School of Nursing, Ann Arbor, MI, USA. [113]University of New South Wales, Department of Psychiatry, Sydney, NSW, AU, USA. [114]Columbia University Medical Center, Department of Medicine, New York, NY, USA. [115]Mental Health Centre Sct. Hans, Institute of Biological Psychiatry, Roskilde, DK, Denmark. [116]Oslo University Hospital, KG Jebsen Centre for Psychosis Research, Norway Division of Mental Health and Addiction, Oslo, NO, USA. [117]University of South Florida College of Public Health, Genomics Program, Tampa, FL, USA. [118]Uniformed Services University, Department of Psychiatry, Bethesda, Maryland, USA. [119]Arq, Psychotrauma Reseach Expert Group, Diemen, NH, Netherlands. [120]Leiden University Medical Center, Department of Psychiatry, Leiden, ZH, NL, Netherlands. [121]Netherlands Defense Department, Research Center, Utrecht, UT, Netherlands. [122]Amsterdam UMC (location VUmc), Department of Anatomy and Neurosciences, Amsterdam, Holland, NL, Netherlands. [123]Amsterdam UMC (location VUmc), Department of Psychiatry, Amsterdam, Holland, NL, Netherlands. [124]Ralph H Johnson VA Medical Center, Department of Mental Health, Charleston, SC, USA. [125]Medical University of South Carolina, Department of Psychiatry and Behavioral Sciences, Charleston, SC, USA. [126]University of Copenhagen, Department of Clinical Medicine, Copenhagen, Denmark. [127]James J Peters VA Medical Center, Department of Mental Health, Bronx, NY, USA. [128]Baylor Scott and White Central Texas, Department of Psychiatry, Temple, TX, USA. [129]CTVHCS, COE for Research on Returning War Veterans, Waco, TX, USA. [130]Yale University, Department of Biostatistics, New Haven, CT, USA. [131]University of Washington, Department of Psychiatry and Behavioral Sciences, Seattle, WA, USA. [132]Harvard School of Public Health, Department of Epidemiology, Boston, MA, USA. *email: cnievergelt@ucsd.edu

