## [Peer Review File · Nature Communications]

Reviewers' Comments:

Reviewer #1:

Remarks to the Author:

The authors have substantially improved their PTSD GWAS paper following the previous sets of reviews (I was reviewer 1). This revision is straightforward to follow, and a pleasure to read - thank you.

The polygenic risk score into MVP is an excellent addition to the paper, and substantially increase's a reader's confidence in the genetic findings.

I have made minor comments below.

Minor comments

1. Introduction "over 20,000 individuals, approximately 25% of whom were cases" It might be clearer for the later comparison with the current study to give the number of cases here.
 2. The standard abbreviation for UK Biobank is UKB (not UKBB)
 3. P9. GWAS. Although the full sample size is $N = 23,212$ cases, as given in the text, only one of the significant findings in Table 1 has the full data set. The other 'all' analysis has only 12k cases, and the male-only analyses. An indication of this in the text, as well as in Table 1 would be useful.
 4. Table 1: The columns of N_{total} , z and SE could be removed to make the table clearer, without losing useful information.
 5. Table 1: What hypothesis is the p-value testing? I assume that h^2 is non-zero for PTSD prevalence of 0.3. Please add to the legend.
 6. Table 2: What does the ** notation refer to?
 7. Table 3: The relevance of the block of text beginning "No significant findings: ..." was not clear to me.
- P12 "Replication of findings in the external MVP cohort"
8. The study looking at replicability across the two cohorts is a nice addition to the manuscript.
 9. A reference for the MVP study should be included here.
10. P12 Polygenic risk scores (PRS) for PTSD. A p_T polygenic risk score p-value threshold is given for the analysis into MVP, but not for the analysis from UKB. Please add.
11. P13. The mtCOJO results are now much clearer to read. I still feel that panels B-E looking at the genetic correlations between other psychiatric disorders distract from the important findings in Panel A.
12. In the Methods, is it possible to separate out the cohort descriptions from the statistical methods?

Reviewer #2:

Remarks to the Author:

The authors have responded adequately to the reviewers' comments. I have no further issues to raise

Reviewer #3:

Remarks to the Author:

As described in prior reviews, I was asked to comment upon the utility of the enclosed findings and their relevance to the broader PTSD research community. I defer to other reviewers for technical &

procedural review. In short, the authors have been responsive to my last round of reviews and associated questions. The findings here are important and highly relevant to the broader community. One remaining item, which is admittedly minor, is that the importance and utility of the polygenic risk score for re-experiencing symptoms in PTSD is over stated. While it is interesting, having a risk score for a single element of a complicated diagnosis provides little meaningful clinical information and should be de-emphasized. The other interpretation is that the lack of a risk score for the majority of symptom clusters tells a more compelling story that the GWAS approach to PTSD still requires significant advances before it can be considered clinically useful.

REVIEWERS' COMMENTS:

Reviewer #1 (Remarks to the Author):

The authors have substantially improved their PTSD GWAS paper following the previous sets of reviews (I was reviewer 1). This revision is straightforward to follow, and a pleasure to read - thank you.

The polygenic risk score into MVP is an excellent addition to the paper, and substantially increase's a reader's confidence in the genetic findings.

I have made minor comments below.

Minor comments

1. Introduction "over 20,000 individuals, approximately 25% of whom were cases" It might be clearer for the later comparison with the current study to give the number of cases here.

- changed to approximately 5,000 (25%)

2. The standard abbreviation for UK Biobank is UKB (not UKBB)

- changed

3. P9. GWAS. Although the full sample size is $N = 23,212$ cases, as given in the text, only one of the significant findings in Table 1 has the full data set. The other 'all' analysis has only 12k cases, and the male-only analyses. An indication of this in the text, as well as in Table 1 would be useful.

- this is a good point; we specified this in the main text and referred to Table 3, that shows the exact number of subjects included for each specific lead-SNP.

4. Table 1: The columns of N_{total} , z and SE could be removed to make the table clearer, without losing useful information.

- removed

5. Table 1: What hypothesis is the p -value testing? I assume that h^2 is non-zero for PTSD prevalence of 0.3. Please add to the legend.

- It is testing if h^2 is different from zero and applies to all prevalences (the adjustments to the liability scale also adjust SE in proportion to the h^2 estimate). We noted this in the legends were applicable.

6. Table 2: What does the ** notation refer to?

- thank you for catching this omission. "not significant when using a stricter Bonferroni correction for sex-split analyses"

7. Table 3: The relevance of the block of text beginning "No significant findings: ..." was not clear to me.

- agreed! we changed this and made it clearer.

P12 "Replication of findings in the external MVP cohort"

8. The study looking at replicability across the two cohorts is a nice addition to the manuscript.

- thank you.

9. A reference for the MVP study should be included here.

- added

10. P12 Polygenic risk scores (PRS) for PTSD. A p_T polygenic risk score p -value threshold is given for the analysis into MVP, but not for the analysis from UKB. Please add.

- both p_T 's were 0.3; we added this information for the UKB in Methods and Results.

11. P13. The mtCOJO results are now much clearer to read. I still feel that panels B-E looking at the genetic correlations between other psychiatric disorders distract from the important findings in Panel A

- Thank you. We decided to keep panels B-E.

12. In the Methods, is it possible to separate out the cohort descriptions from the statistical methods?

- Yes, the individual study descriptions have been moved to the SI file as "Supplementary Methods"

Reviewer #2 (Remarks to the Author):

The authors have responded adequately to the reviewers' comments. I have no further issues to raise

- we thank the reviewer for his comments.

Reviewer #3 (Remarks to the Author):

As described in prior reviews, I was asked to comment upon the utility of the enclosed findings and their relevance to the broader PTSD research community. I defer to other reviewers for technical & procedural review. In short, the authors have been responsive to my last round of reviews and associated questions. The findings here are important and highly relevant to the broader community. One remaining item, which is admittedly minor, is that the importance and utility of the polygenic risk score for re-experiencing symptoms in PTSD is over stated. While it is interesting, having a risk score for a single element of a complicated diagnosis provides little meaningful clinical information and should be de-emphasized. The other interpretation is that the lack of a risk score for the majority of symptom clusters tells a more compelling story that the GWAS approach to PTSD still requires significant advances before it can be considered clinically useful.

- We agree with the reviewer and incorporated this comment. "However, larger sample sizes are needed to achieve sensitivity and specificity at levels of clinical utility."